# Teaching Transformers to Solve Combinatorial Problems through Efficient Trial & Error

**Panagiotis Giannoulis**[1]    **Yorgos Pantis**[2,3]    **Christos Tzamos**[2,3]

[1] National Technical University of Athens, Greece
[2] National and Kapodistrian University of Athens, Greece
[3] Archimedes, Athena Research Center, Greece

## Abstract

Despite their proficiency in various language tasks, Large Language Models (LLMs) struggle with combinatorial problems like Satisfiability, Traveling Salesman Problem, or even basic arithmetic. We address this gap through a novel trial & error approach for solving problems in the class NP, where candidate solutions are iteratively generated and efficiently validated using verifiers. We focus on the paradigmatic task of Sudoku and achieve state-of-the-art accuracy (99%) compared to prior neuro-symbolic approaches. Unlike prior work that used custom architectures, our method employs a vanilla decoder-only Transformer (GPT-2) without external tools or function calling. Our method integrates imitation learning of simple Sudoku rules with an explicit Depth-First Search (DFS) exploration strategy involving informed guessing and backtracking. Moving beyond imitation learning, we seek to minimize the number of guesses until reaching a solution. This is achieved using depth-1 guessing, showing empirically that almost all Sudoku can be solved using the puzzle's rules with at most one guess. We provide a rigorous analysis of this setup formalizing its connection to a contextual variant of *Min–Sum Set Cover*, a well-studied problem in algorithms and stochastic optimization.

## 1   Introduction

Large Language Models (LLMs) based on the Transformer architecture have shown remarkable performance in a wide range of Machine Learning tasks, achieving state-of-the-art results on language understanding and generation benchmarks [VSP$^+$17, BMR$^+$20, RWC$^+$19]. Trained to predict the next token in large text corpora, these models acquire strong reasoning abilities, but whether they perform genuine logical reasoning or rely on pattern-matching remains under debate [VMO$^+$23, DLS$^+$23]. Prompting strategies such as Chain-of-Thought and its many variants [NAGA$^+$21, WWS$^+$22, YYZ$^+$23, WWS$^+$23, BBK$^+$24, CMWC23, KTF$^+$22, ZSH$^+$23, MTG$^+$23] have enabled LLMs to exhibit multi-step reasoning capabilities and solve even complex mathematical problems [HBK$^+$21]. Yet, LLMs still struggle in more structured domains like arithmetic and combinatorial optimization. Controlled evaluations on synthetic tasks, including algorithmic puzzles, formal grammars, logic-based games, and arithmetic problems, have revealed consistent failure modes of current models and methods [LAG$^+$22, LHB$^+$23, NLW23, AZL23, YXLAZ24a, YXLAZ24b, LSL$^+$24]. At the same time, the literature highlights that Transformer-based models have the potential to leverage

---

Authors listed alphabetically; equal contribution.

Code available at `https://github.com/gpt-reasoning/ReasoningCombinatorials`

Table 1: Performance of five state-of-the-art LLMs on solving Sudoku puzzles in 9×9 format. The first two are general-purpose LLMs, while the remaining three are reasoning-focused models.

| Model | Board Accuracy | Cell Accuracy | Time (sec./puzzle) |
|---|---|---|---|
| GPT–4o | 0.00% | 44.62% | 1.60 |
| Gemini–1.5 Pro | 0.00% | 31.70% | 3.20 |
| GPT–o3 mini | 0.00% | 30.99% | 265.70 |
| Gemini–2.5 Flash | 0.00% | 20.00% | 250.80 |
| DeepSeek–R1 | 0.00% | 41.48% | 510.40 |

internal mechanisms like attention to capture combinatorial structure without specialized heuristics or external tools. However, appropriately training them to display this behavior remains a major challenge and an active topic of research in robustness and generalization.

In this work, we focus on improving reasoning in LLM-style Transformers for combinatorial problems. We develop a framework for teaching Transformers to identify a path towards a solution to a combinatorial problem through effectively exploring different alternatives with trial-and-error. Our framework relies on training the model to distinguish valid and invalid solutions and incrementally builds a solution with clever guesses. This approach applies in very general settings and can be used to solve any combinatorial problem that belong to the NP-class, like Satisfiability, Clique detection or the Traveling Salesman Problem, where verifying a valid solution can be done efficiently.

Our primary testbed in this work is Sudoku, chosen for its well-defined rules, extensively studied difficulty levels, and widespread use as a reasoning benchmark. A standard Sudoku puzzle consists of a $9 \times 9$ grid, subdivided into nine $3 \times 3$ boxes. The grid is partially filled with digits from 1 to 9, and the goal is to complete it subject to the following rules: (i) each cell contains exactly one digit, (ii) each row contains every digit exactly once, (iii) each column contains every digit exactly once, and (iv) each box contains every digit exactly once. We focus on the standard Sudoku puzzles, while our approach can be directly applied to its generalized form with $n \times n$ grids which is NP-complete [YS03] and thus equivalent to any problem in the class NP.

Solving Sudoku puzzles without external tools is extremely challenging even for state-of-the-art industrial-scale LLMs. We evaluated several models, including OpenAI GPT [AAA+23], Google Gemini [TAB+23], and DeepSeek [GYZ+25], on randomly generated Sudoku puzzles. As shown in Table 1, none of these models were able to produce correct solutions. We tested both general-purpose LLMs and reasoning-focused variants, since Sudoku demands logical reasoning. General-purpose models generated answers quickly, whereas reasoning-oriented models ran significantly longer as they attempted step-by-step strategies mirroring human solving patterns. However, once they made an incorrect inference, these models could not backtrack and revise their solutions, often ending with contradictory board states. Additional details are provided in Appendix B.

Beyond these general-purpose models, several methods have been proposed in the literature to solve Sudoku puzzles via Deep Learning although often relying on specialized architectures. [MKPZ11] investigates a Neural Network (NN)-based approach based on Hopfield Networks as an integer optimization tool, addressing its convergence limitations through a coprocessor NN designed to solve linear programming problems. [PPW18] introduces the Recurrent Relational Network (RRN), a generalization of the Relational Network [SRB+17], to evaluate multi-step relational reasoning. Their method involves multiple rounds of message passing between Sudoku puzzle cells to arrive at a solution. [YIL23] explores solving Sudoku puzzles using a Recurrent Transformer model, integrating Sudoku's constraints directly into both the model architecture and the training process.

While effective in solving even difficult puzzles, these prior models diverge from the LLM paradigm, and their architectural customizations limit the transferability of their insights to combinatorial reasoning with LLMs. Moreover, these methods follow a direct solution approach: given the initial state of the puzzle, the network attempts to predict the entire solution in a single forward pass. This design introduces critical limitations. First, there is no mechanism for error recovery, so if the model generates an incorrect solution, it cannot easily revise or correct its output. Second, the reasoning process remains opaque. Unlike humans, who solve puzzles step-by-step, these models do not expose their intermediate decision-making, resulting in behavior that is difficult to interpret or trust.

Closer to our work, [SDWP24] study how a causal language model (GPT-based) can reliably solve Sudoku puzzles. By reframing Sudoku as a next-token prediction task over transcripts of human-like reasoning based on known Sudoku strategies, their Causal Transformer learns to imitate and apply these heuristics incrementally, typically executing simpler strategies before tackling more complex ones. This approach showcases the promising capability of Autoregressive Transformers to successfully internalize and deploy multi-step reasoning procedures in a more interpretable way. However, their proposed method trains the Transformer on a structured sequence of logical steps, which do not allow the model to make mistakes. This inherently limits its problem-solving capability to strategies present in the training data whereas a large fraction of puzzles are not solvable by simple logic steps. As such, its performance is significantly worse than previous task-specific models and even for puzzles solvable by logic, the model is still not perfect potentially due to the complexity of applying the given chain of heuristics.

Another line of works investigates the reasoning capabilities of Masked Diffusion Models (MDMs) in language generation through the Sudoku task. [YGG+25] show the potential of MDMs to outperform autoregressive models in reasoning and planning with fewer parameters, highlighting their potential impact on downstream applications. [KSK+25] investigate the same models, exploring sampling strategies and token ordering effects, and propose an adaptive inference method that further improves performance. While [SDWP24] is more closely related to our work, our results are also comparable to those of [KSK+25], who outperformed [YGG+25]. This demonstrates that our methodology surpasses all existing state-of-the-art approaches for solving combinatorial tasks such as Sudoku.

## 2   Methodology

In this work, we introduce a novel framework for reasoning with Transformers. Our method combines imitation learning with trial-and-error search, enabling models to go beyond learned heuristics. For combinatorial problems such as Sudoku, a verifier exists that can check whether a partial or complete assignment satisfies the problem's rules; we therefore use this verifier to validate the correctness of each attempted instance. Specifically, when no further logical steps can be applied, the model is trained to make informed guesses, just as humans do. Crucially, it can backtrack when a guess eventually leads to a dead end, and attempt alternative continuations in a Depth-First Search (DFS) manner. By coupling simple learned rules with the flexibility to explore, while continuously relying on the verifier for correctness checks, our method navigates the solution space efficiently. The resulting Causal Transformer solves puzzles robustly and interpretably, showcasing emerging behaviors of progressive reasoning and self-correction.

Our experiments showcase the capability of Transformers to learn this algorithmic behavior. Specifically, we observe that the models effectively learn to fill the values based on the Sudoku rules.

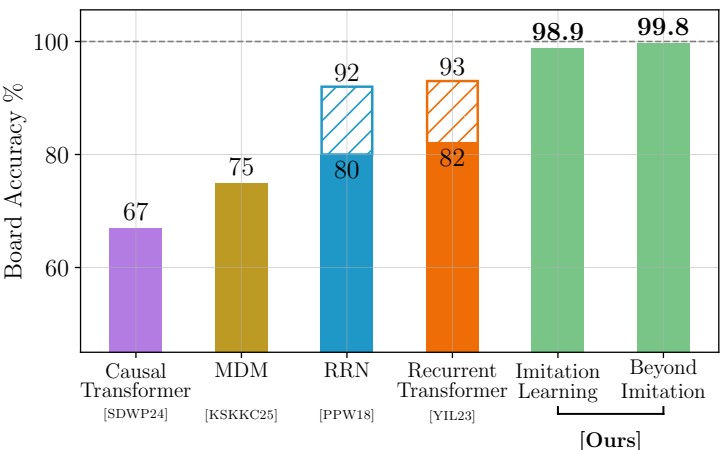

Figure 1: Comparison of board accuracy with previous state-of-the-art models on 100K randomly generated Sudoku. As some models are trained on a different dataset, we retrain them using our random Sudoku generator and report their increased accuracy with shaded bars.

When direct rule application is exhausted, the model guesses a candidate value and recovers correctly from incorrect guesses by backtracking when a dead end is identified, as in the DPLL algorithm [DP60, DLL62]. As a result of these learned abilities, our method achieved near-perfect accuracy on Sudoku puzzles (99%), demonstrating state-of-the-art performance for a NN-based Sudoku solver (see Figure 1).

Our framework applies to any NP problem, whose solutions are hard to find but easy to verify. To demonstrate its generality, we also apply it to the canonical NP-complete problem of Satisfiability [Coo71], specifically the 1-in-3 SAT variant, where each clause contains three literals and exactly one must be true. Even in this task, our trained Transformer is able to correctly find solutions 99% of the time. Note that 1-in-3 SAT is a canonical NP-complete problem, and by the Cook-Levin theorem [Coo71], any problem in NP can be written as 1-in-3 SAT. However, some problems may benefit from problem-specific optimizations and therefore, we provide a library in Appendix F.2 that lets one generate transcripts for any problem in NP by providing the appropriate components.

A key ingredient for our framework is generating appropriate training transcripts via a DFS routine that guesses partial assignments, checks them for correctness, and either recurses or backtracks as needed. We represent the problem-solving process as a sequence of discrete actions and use an action-level tokenization. This allows us to use a multiple-target approach during training, where the output of each token is treated as a distribution of probabilities over the next possible actions (soft-labels), enabling richer learning signals compared to single-target approaches. We provide an efficient Python library (written in C) for generating the transcripts and the corresponding targets (see Appendix F.2). The library can be easily extended to provide reasoning transcripts for any combinatorial problem.

**Beyond imitation learning.** Whereas earlier studies struggled with accurately solving Sudoku, our approach can teach Transformers to solve combinatorial problems with near-perfect accuracy. Our thesis is that once LLMs can learn to solve different problems correctly, the next milestone is to also minimize the reasoning time. In fact, it is well-known that any combinatorial problem can be solved given sufficient time. As long as Transformers can reliably detect valid solutions to a problem, they can also solve it by randomly generating tokens and checking whether the produced transcript contains a solution. According to the "infinite-monkey" theorem [Mon], an infinitely large sequence of random tokens will almost-surely contain a valid solution to the problem instance.

For the second part of our work, we aim to use the predictive abilities of Machine Learning to efficiently navigate the space of solutions with the goal of minimizing the expected number of reasoning steps. While in the first part, our goal was to imitate how humans solve problems systematically, in this stage we aim to do even better. Humans themselves are limited in their ability to solve combinatorial problems and the DFS method outlined above only works for small instances; it quickly becomes intractable as problem size increases. In fact, we do not know of efficient methods for solving most problems in the class NP which is reflected by the major open problem of P vs NP. An important promise of using Machine Learning for combinatorial problems is to develop data-driven methods for solving problems more efficiently than hand-crafted methods can.

Minimizing the time to solution is known as the *min-sum* objective in combinatorial optimization. To perform a theoretically grounded analysis of this setting, we make two important assumptions: (i) *one-level guessing*, where we only allow the method to make a single guess and any second guess counts as failure, and (ii) *non-adaptive guessing*, where for every new guess point we proceed without retaining any memory of previous attempts. Under these restrictions, any instance solvable with one guess will eventually be solved in finite expected time. Remarkably, on randomly generated Sudoku boards, a single guess suffices to correctly solve ∼99.8% of them: after applying the standard set of simple rules, nearly every board admits a *backdoor* i.e. a guess that, when followed by those same rules, completes the puzzle. In this setup, optimizing the probability of correctly guessing the backdoors directly corresponds to minimizing the solution length. We formulate this problem as a contextual variant of the MIN-SUM SET COVER, a problem commonly studied in algorithms and stochastic optimization (see Appendix E). Through this connection, we identify a novel loss function that directly captures the length of the solution and outperforms the standard Cross-Entropy loss, both theoretically and empirically. Our experiments show that for the Sudoku task the solution lengths of the trained model compare favorably to those of an oracle that knows the correct values of all remaining cells without knowing which are backdoors.

Despite its simplicity, this setup remains surprisingly effective, *even outperforming the previous multi-level guessing method*. This advantage arises because, in our beyond imitation learning approach, training uses only successful trajectories, i.e., those that contain the correct guess. While imitation learning provides richer supervision by including both correct and incorrect guess tokens, it also encourages the model through next-token prediction to continue generating incorrect tokens, rather than identifying the correct path. By excluding incorrect tokens from the training data, our approach goes beyond imitation learning and concentrates learning on high-value decisions that directly lead to a solution.

**Experimental setup and puzzle generation.** All our experiments follow a Causal Transformer (GPT-2 variant [RWC$^+$19]) that is identical to that of [SDWP24] to allow for a direct comparison. The model comprises 8 layers, each with 8 self-attention heads, an embedding dimension of 576, and a feed-forward network with hidden dimension 3456 ($6 \times$ the embedding size). The model also employs causal masking within its attention layers to prevent access to future tokens. In total, this model contains 42M parameters.

We train and evaluate our model using randomly generated Sudoku. We deviate from previous works that used pre-existing datasets, such as the one provided by [Rad20], striving for a clean and interpretable distribution of Sudoku puzzles. In particular, our generator works exactly as follows:

> Out of the 6,670,903,752,021,072,936,960 distinct Sudoku boards that exist, it picks one uniformly at random and removes entries based on a uniformly random permutation of the 81 cells, skipping an entry if its removal yields $> 1$ solution.

This process results in a nearly identical distribution of puzzles to the Kaggle dataset used in previous work but arises in a very principled way. Relying solely on static datasets can introduce bias into the evaluation, as these datasets may represent only a selective portion of the true distribution. Moreover, having a generator instead of a dataset is more appropriate for combinatorial problems allowing stream-like training and avoiding overfitting issues as every puzzle is only seen once. We release an efficient generator as a Python library (written in C), named *SudokuPy*[1], but the same distribution can also be reproduced using the description above. Full functionality details appear in Appendix F along with a description of the random generator for random instances of 1-in-3 SAT that we use.

**Summary of our contributions.** Our key contributions can be summarized as follows:

**A novel trial-and-error reasoning framework.** We introduce a principled trial-and-error reasoning framework for solving combinatorial puzzles using LLM-style Causal Transformers. This framework leverages two key ingredients:
(i) Action-Level Tokenized Transcript Generation: A method for representing the problem-solving process as a sequence of discrete actions. Importantly, during training, we utilize a multiple-target approach, where the output of each token is treated as a distribution of probabilities over the next possible actions (soft-labels), enabling richer learning signals compared to single-target approaches.
(ii) Exploration Beyond Heuristics: The ability to explore the solution space through informed guessing and backtracking, going beyond reliance on handcrafted heuristics.

**State-of-the-art performance on challenging combinatorial tasks.** Our LLM-style architecture achieves a near-perfect accuracy of 99% on Sudoku puzzles, surpassing previous NN methods. Furthermore, we attain 99% accuracy on the 1-in-3 SAT task showcasing the framework's versatility (see Appendix G).

**Principled optimization of the guessing process.** We theoretically analyze and optimize the crucial guessing component of our framework by reframing it as a contextual MIN-SUM SET COVER problem. Through this connection, we identify a novel loss function that directly captures the length of the solution and outperforms the standard Cross-Entropy loss, both theoretically and empirically.

**A standardized benchmark with an efficient puzzle generation library.** To facilitate future research, we introduce a fast and lightweight Python library for generating uniformly random Sudoku puzzles from the entire solution space. This enables streaming-like training and provides a standardized benchmark for evaluating and comparing different reasoning models.

---

[1]Code available at `https://github.com/gpt-reasoning/sudokupy`

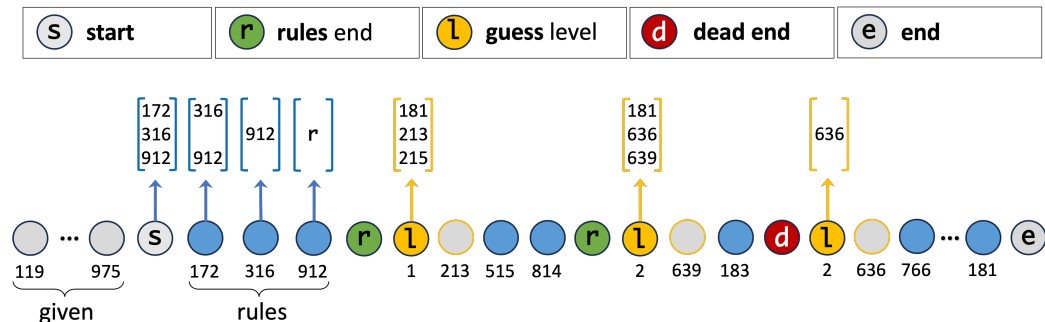

Figure 2: Example of a training transcript. Values in blue brackets indicate the multiple valid labels for the output of each token during the rule-application step. Yellow brackets show the set of valid candidates when the model reaches a guess token at a given level. The selected guess is shown as a gray circle with yellow outline. If it leads to a dead end, a follow-up guess is made at the same cell.

## 3 Imitation learning – DFS algorithm

### 3.1 Method overview

**Baseline approach and rule logic.** In [SDWP24], the authors train a GPT-2 variant to solve Sudoku puzzles using supervised imitation learning. Their training data consists of step-by-step solution transcripts that progressively apply seven human-crafted strategies (namely the Lone Single, Hidden Single, Naked Pair, Naked Triplet, Locked Candidate, XY Wing, and Unique Rectangle), ordered from simpler to more complex. For token encoding, each cell is represented using three tokens: $(r, c, v)$, where $r$ and $c$ denote the cell's row and column indices, and $v$ is the digit placed at that position.

In our approach, rather than incorporating complex human-designed strategies into the training transcripts, we rely solely on applying the four fundamental Sudoku rules. These essentially correspond to the heuristics Lone Single and Hidden Single. Specifically, during rule-based inference, we fill cell values according to the following constraints: (i) if an unfilled cell $(r, c)$ can only be filled with one value, fill it, (ii) if in a row $r$ the digit $v$ can only be filled in a single position, fill it, (iii) if in a column $c$ the digit $v$ can only be filled in a single position, fill it, (iv) if in a $3 \times 3$ box the digit $v$ can only be filled in a single position, fill it.

**Encoding choice.** We adopt a novel tokenization strategy, encoding each move as a single 3-digit number ranging from 111 to 999, with the first digit representing the row r the second the column c and the third the value v. This differs from [SDWP24], that use 3 separate tokens for the row, column and value. Our action-level representation reduces the input sequence length by a factor of 3, leading to faster training iterations and more efficient solution generation during evaluation.

**Multiple targets.** In standard language modeling tasks, each training example consists of a fixed sentence, and the model is trained to predict a single next token at each step. In contrast, combinatorial puzzle transcripts often permit multiple valid next moves at any given point. To leverage this property, we modify the loss function to support multiple next-token predictions, rather than assuming a single deterministic target. Specifically, instead of using the standard Cross-Entropy loss $-\log p_i$, we use the sum of Cross-Entropies across all valid targets: $-\sum_{i \in S} \log p_i$, where $S$ denotes the set of all valid next tokens at that step. This enables the model to treat one input as having multiple plausible continuations, enriching its exposure to diverse sequence paths during training. Our experiments demonstrate that this adjustment facilitates faster learning of the task (see Figure 3).

**Algorithm.** Our algorithm is trained using transcription sequences (see Figure 2) that proceed through the following steps. The input begins with the sequence of moves (tokens) corresponding to the initially filled cells, followed by a "start" token $s$. Given this input, the model first attempts to fill all resolvable cells by iteratively applying the basic Sudoku rules. If the board is completed during this phase, the model outputs the "end" token $e$. Otherwise, it outputs a "rules end" token $r$, indicating that no further cells can be filled using rules alone and that guessing is required.

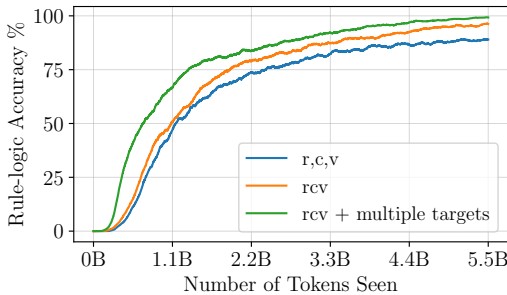
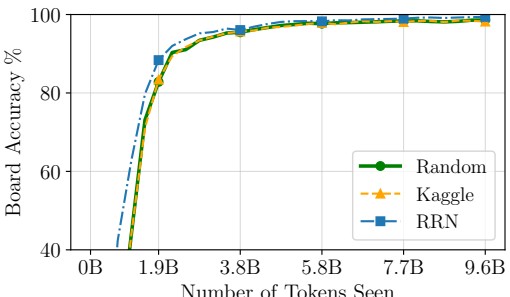

Figure 3: Rule-logic accuracy during training for three model variants differing in token encoding and loss function.

Figure 4: Board accuracy during training on samples from our generator, evaluated on Random, Kaggle unfiltered, and RRN datasets.

At this point, the model outputs the current "guess level" token (an integer from 1 to 99 representing the number of active guesses) and proceeds to select an informed guess from the remaining valid candidates. This guessed move becomes the next token. The model then re-enters the rule-application phase and fills in as many cells as possible based on the updated board.

If unresolved cells remain, the model increases the guess level and repeats the guess-and-fill process. In the event of a conflict (e.g., no valid values exist for a cell in the current state of the board), the model outputs a "dead end" token $d$ to signal failure at the current search path. It then backtracks to the last guess level and selects a new guess different from the one that led to the conflict. This guess is made on the same cell by selecting a different potential value.

This combination of rule application, informed guessing, and backtracking continues recursively with a DFS approach until a valid solution is found or a maximum-allowed sequence length is reached. At inference, we get the solution choosing for any cell its last assigned value in the generated sequence.

## 3.2 Benchmarks

We evaluate our model across four distinct datasets to benchmark its performance and ensure generalization: (i) Random: our own puzzle generator, used to produce both training data and a held-out 100K test set, (ii) Kaggle unfiltered: the Kaggle dataset [Rad20], a uniformly sampled collection of 3M puzzles, from which we randomly sample 100K for evaluation, (iii) Kaggle filtered: a 1.9M puzzle subset of the unfiltered Kaggle dataset, curated by [SDWP24] to include only puzzles solvable using seven Sudoku strategies, and (iv) RRN: the dataset introduced by [PPW18], consisting of 180K training and an 18K test set, stratified by clue count (ranging from 17 to 34 initial givens).

## 3.3 Results

We begin by evaluating how effectively the Transformer learns to apply basic Sudoku rules, and examine how our design choices, specifically the action-level token encoding and multi-target loss, affect the speed and efficiency of learning. In this setting, rule-logic accuracy is defined as the model's ability to correctly fill all cells resolvable by rule-based inference. If even a single such cell is left unfilled or incorrectly filled, the entire board is considered incorrect. Figure 3 shows the evolution of rule-logic accuracy during training on our Random dataset, plotted against the number of tokens seen (in billions), which also correlates closely with training time. We compare three configurations: (i) the baseline setup from [SDWP24] using their triplet tokenization and single-target loss, (ii) our compact encoding with single-target loss, and (iii) our full approach with both compact encoding and multi-target loss. While all configurations eventually achieve high accuracy, our method substantially accelerates learning, with the multi-target variant reaching near-perfect performance much earlier.

We then track the evolution of board accuracy throughout training on the full Sudoku-solving task, across three test sets: Random, Kaggle unfiltered, and RRN. As shown in Figure 4, our generated puzzles yield board accuracy curves that closely mirror those from the Kaggle set, rising in parallel throughout training. This suggests that our generator samples uniformly from the puzzle space, similar to the Kaggle dataset, while also offering the advantages of stream-like training and full

Table 2: Board accuracy comparison of neural Sudoku solvers across training and test sets.

| Method | Train Data | Test Data | | | |
|---|---|---|---|---|---|
| | | Random | Kaggle unfiltered | Kaggle filtered | RRN |
| RRN [PPW18] | RRN | 79.30% | 79.60% | 90.70% | 98.90% |
| | Random | 92.40% | 92.40% | 98.30% | 97.90% |
| Recurrent Transformer [YIL23] | RRN | 82.20% | 82.10% | 93.90% | **99.50%** |
| | Random | 92.60% | 92.50% | 99.30% | 99.00% |
| Causal Transformer [SDWP24] | Kaggle filt. | 67.16%[†] | 67.16% | 87.18% | 94.80% |
| MDM [KSK⁺25] | Kaggle filt. | 74.97%[†] | 74.97% | 89.49% | -[‡] |
| **Causal Transformer Trial & Error (ours)** | Random | **98.90%** | **98.90%** | **99.50%** | 99.40% |

[†]Accuracy approximated from Kaggle unfiltered performance due to dataset similarity.
[‡]Accuracy is unreported because the model code has not been released.

distribution coverage. In contrast, the RRN dataset appears easier to solve in practice, with board accuracy rising more quickly, which indicates that its puzzles may be biased toward simpler instances.

Regarding training, our model is trained using puzzles generated by our random generator. For the baselines, the Causal Transformer from [SDWP24] was trained on the Kaggle filtered dataset; the same holds for MDM [KSK⁺25]. The RRN [PPW18] and Recurrent Transformer [YIL23] models were originally trained on the RRN dataset. For a fair comparison, we also trained these models on our random generator using their released training code.

The following key takeaways emerge from the comparative analysis in Table 2: first, our trial-and-error Transformer achieves significant improvements over the Causal Transformer baseline [SDWP24], with absolute accuracy gains of 31.7% (Kaggle unfiltered), 12.3% (Kaggle filtered), and 4.6% (RRN). Notably, while their model is restricted to puzzles solvable by seven human-crafted strategies (covering only 1.9M of Kaggle's 3M puzzles), our method generalizes across the full distribution, achieving 98.9% accuracy on all 3M puzzles. Also, while domain-specific models like RRN and the Recurrent Transformer achieve strong performance on the RRN benchmark, our method outperforms them by 6-7% on the more diverse Random and Kaggle datasets, even when these competitors are enhanced by training on our random generator. For example, our model achieves 98.9% vs. their 92.4-92.6% on Random dataset. Finally we see that training on our random generator improves performance on diverse datasets (e.g., RRN's +13.1% gain on Random test) while preserving RRN benchmark accuracy (97.9% vs. 98.9% when RRN-trained), demonstrating broad coverage without compromising specialization.

## 4   Beyond imitation learning – optimizing guesses

In our previous approach, we used imitation learning to teach Transformers to solve combinatorial problems, achieving near-perfect accuracy. Having reached this milestone, we now shift our focus from correctness to efficiency. Our goal is to leverage the predictive power of Transformers to minimize the expected reasoning time, measured by the number of steps needed to reach a valid solution. To this end, we focus on optimizing the guessing steps, as they constitute the primary source of variability in solution length.

To study this systematically and connect with theoretical foundations, we study a simplified variant of our setup. We impose two constraints: (i) depth-1 guessing, where only a single guess is allowed before restarting; and (ii) non-adaptive policies, where each failed attempt triggers a full restart with no memory of past guesses. This reduced problem closely resembles the MIN-SUM SET COVER problem, where the objective is to choose elements (guesses) that cover (solve) the problem as early as possible in expectation. We analyze theoretically the connection in Appendix D, where we quantify the performance degradation of non-adaptive methods compared to adaptive ones.

Despite its simplicity, the setup remains surprisingly effective: on randomly generated Sudoku boards, our policy achieves **99.8% accuracy** on instances from our random generator; *even beating the previous multi-level guessing method*. This means that 99.8% of random Sudoku can be solved by

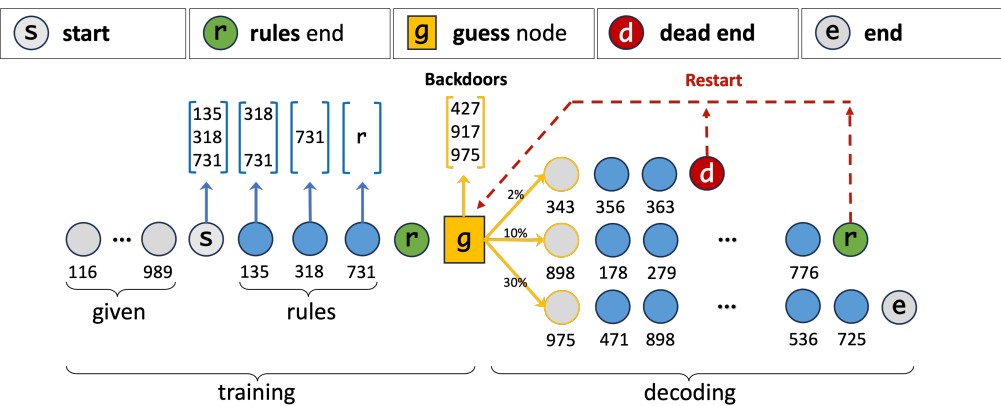

Figure 5: One-level guessing with restarts. The figure illustrates both training and decoding. During decoding, a guess is sampled from the output of the guess node. If the guess is not a backdoor, even if correct, subsequent application of rules fails to produce a solution and a restart is triggered.

first applying a fixed set of simple rules until only a single backdoor guess is required; once that guess is made, the same simple rules are sufficient to complete the solution. This property, firstly uncovered in our work, and is of independent interest. Note that backdoor move refers to such a single guess such that once it is made, the same simple rules can complete the entire solution. Identifying the correct backdoor is non-trivial. Since our model does not know in advance which cell is the backdoor, it often requires multiple guesses before finding the right one (see Figure 6).

Our method is illustrated in Figure 5. Unlike our earlier deterministic (argmax) generation policy, here we sample guesses from the model's token distribution at the guess node. If the sampled guess leads to a full solution via rule application, the process terminates; otherwise, it restarts. Crucially, only successful trajectories, those containing the correct guess, are used for training. This design makes the approach more efficient than our previous imitation learning setup: while imitation learning includes incorrect guess tokens and thus provides richer supervision, it also encourages the model to learn to predict subsequent wrong tokens via next-token prediction, rather than immediately identifying the correct path. By excluding incorrect guesses from the training data, this approach goes beyond imitation learning focuses learning on high-value decisions. The guess node, viewed as a single decision step, may represent either a single action/token or a thinking sequence (see Appendix D). This process continues until a valid trajectory terminates at an "end" token, indicating completion. This framework offers a tractable yet powerful setting to study how well LLMs can prioritize high-value guesses under uncertainty.

Central to our optimization is the training objective applied at the output of the guess node, which must encourage the model to concentrate probability mass on all valid backdoor moves. Specifically, if $S$ is the set of valid backdoor moves at a given guess node, and $p_i$ denotes the probability assigned to each such move, we consider different losses for optimization. The loss corresponding to the expected number of trials is $\mathcal{L}_1 = (\sum_{i \in S} p_i)^{-1}$ whereas the standard Cross-Entropy loss with multiple targets is $\mathcal{L}_2 = -\sum_{i \in S} \log p_i$. For numerical stability during training, we minimize $\log \mathcal{L}_1$, implemented using the log-sum-exp trick. One can interpret the $\mathcal{L}_1$ loss in terms of a geometric distribution: we treat each good guess as a Bernoulli trial where a "success" means finding a valid backdoor. Not every correct guess actually works as a backdoor, so each attempt can succeed or fail. The goal is to minimize the expected number of trials needed to find a working backdoor; analogous to minimizing the expectation in the geometric distribution.

In Appendix E, we theoretically compare the two losses $\mathcal{L}_1$ and $\mathcal{L}_2$ showing that $\mathcal{L}_1$ is theoretically optimal whereas $\mathcal{L}_2$ may result in excessively long sequences. Figure 6 compares empirically the performance of the two loss functions, $\mathcal{L}_1$ and $\mathcal{L}_2$, against two reference baselines using a cumulative histogram. The x-axis indicates the number of guesses required to reach a valid solution, while the y-axis shows the percentage of instances solved within that number of guesses. To interpret the results, we compare our method against two theoretical oracles: a) upper-bound oracle and b) lower-bound oracle. The upper-bound oracle corresponds to knowing the full Sudoku solution but not which cells are backdoors. This corresponds to guessing the correct value for a random unfilled

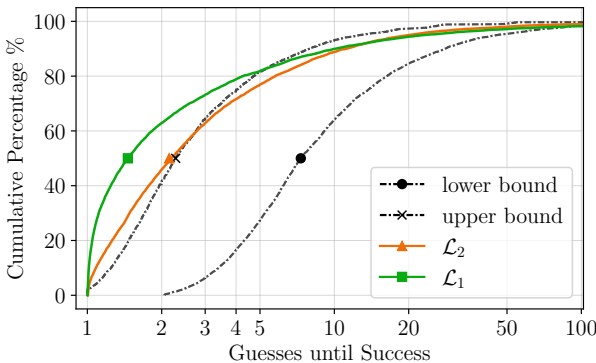

Figure 6: Cumulative histogram of guesses needed to solve Sudoku instances for two losses and two baselines. The lower bound corresponds to random guessing among the remaining candidates for all unfilled cells, while the upper bound corresponds to random guessing the correct value among all unfilled cells without knowing which are backdoors. Remarkably more than 80% of the puzzles are solved with fewer guesses than the oracle upper bound.

cell (after the basic rules are applied) but this guess may not be a backdoor resulting in a restart. The lower-bound is similar but does not assume we know the solution. After the basic rules are applied, it chooses a uniformly random (cell, value) combination that does not conflict with the existing values, i.e. appears valid but may not be correct and restarts if the combination is not a backdoor. Note that a backdoor must necessarily be correct as the solution to a Sudoku is always unique. Both oracles serve only as reference curves, one optimistic (upper bound) and one pessimistic (lower bound), to help contextualize the results. They are purely theoretical and not tied to transformers or any specific model architecture.

We observe that both of our loss functions significantly outperform the lower bound. Notably, the theoretically optimal loss $\mathcal{L}_1$ achieves the best performance, especially in the low-guess regime, with a steeper curve and a median of just 1.5 guesses. The sum of Cross-Entropies loss $\mathcal{L}_2$ also performs well, closely matching the upper bound and achieving a median of 2.2 guesses compared to 2.3 for the upper-bound baseline. These results confirm that both losses guide the model effectively, while $\mathcal{L}_1$ offers a meaningful advantage in reducing the expected number of trials.

## 5 Summary, limitations and future work

In this work, we demonstrate that Transformers can be trained to solve combinatorial problems by effectively exploring solution paths through trial and error. Starting with imitation learning, we teach the model to make guesses and backtrack when simple rules are not sufficient which leads to nearly perfect accuracy of 99% on the Sudoku and the 1-in-3 SAT tasks. Beyond imitation learning, we introduce a principled optimization strategy based on the MIN-SUM SET COVER problem. This approach addresses the inefficiencies of naive guessing, resulting in optimized inference time. For the Sudoku task it even results in higher board accuracy of 99.8%. Our approach is directly compatible with any LLM training pipeline as it is based on standard causal Transformers and only requires using our appropriate transcript sequences and corresponding multi-target losses. We hope that these ideas will enable increased performance of LLMs in tackling combinatorial tasks.

While very effective for the Sudoku task, our method for optimizing the number of guesses is limited to depth-1 and non-adaptive policies. Extending the optimization beyond these two constraints is an interesting direction for further research. Even with non-adaptivity, we can still consider randomized multi-level policies as performing a random walk in the search tree of guesses until a valid solution is reached. Optimizing over such policies would automatically identify how deep the search should go, which guess to make next and when to backtrack even if no dead-end has been identified. While the architecture could directly follow that of Figure 5 restarting to previous guess nodes if the model chooses so, training the weights appropriately requires carefully thinking about the corresponding loss functions. This is an interesting challenge to explore both theoretically and empirically.

## Acknowledgments and disclosure of funding

This work has been partially supported by project MIS 5154714 of the National Recovery and Resilience Plan Greece 2.0 funded by the European Union under the NextGenerationEU Program. AWS resources were provided by the National Infrastructures for Research and Technology GRNET and funded by the EU Recovery and Resiliency Facility.

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

# A    Related work

**Sudoku.**    Sudoku puzzles are played on an $n^2 \times n^2$ grid divided into $n \times n$ blocks. The standard version has a rank of 3, resulting in a $9 \times 9$ grid. Solving a Sudoku puzzle is an NP-complete problem [YS03], meaning that while verifying a solution is computationally efficient, finding one is generally hard. Although there are more than $6.67 \times 10^{21}$ possible Sudoku grids, only approximately 5.47B are essentially distinct after accounting for symmetries. A valid Sudoku puzzle must have a unique solution. It has been proven through exhaustive computational analysis [MTC14] that at least 17 clues are required to ensure a unique solution. Further insights into Sudoku, its mathematical properties, and solving techniques can be found in related works [FJ06, RJ06].

Beyond the traditional text-based representation of Sudoku, as described in the main body, several studies have approached the problem from a visual perspective. These models, often referred to as neural-symbolic (NeSy) systems, perform constraint satisfaction by reasoning over their outputs to produce structured predictions. Typically, this involves some form of joint reasoning that integrates prior knowledge and constraints directly into the prediction process. SATNet [WDWK19] is a differentiable MAXSAT solver that employs semi-definite program relaxations to solve MAXSAT problems. It can be integrated as a layer within Deep NNs to tackle composite learning tasks that combine visual perception with logical reasoning, such as solving hand-written visual Sudoku puzzles. SATNet is notable for its ability to learn to solve visual Sudoku without relying on hand-crafted rules or symbolic supervision. However, subsequent work by [CFLS20] identified a critical flaw in the experimental setup: a label leakage issue that inadvertently exposed the model to information about the correct outputs during training. This issue was later addressed by [TRS21], who introduced a self-supervised clustering and distillation process for training a visual classifier within the SATNet architecture. Their approach uses self-supervised pre-training to solve visual Sudoku, effectively

addressing the Symbol Grounding Problem observed in the original SATNet method. Recent research continues to explore visual representations of Sudoku and to develop new NeSy models. Examples include the work of [APD+22, PDA+23, MAB+23, vKTT+23, MMMG24]. Recently, another line of work has explored the visual representations of Sudoku using Spatial Reasoning Models, a variation of diffusion/flow models, demonstrating their ability to solve Sudoku [WPSL25, PWSL25].

**MIN-SUM SET COVER problem.** The MIN-SUM SET COVER problem is a fundamental combinatorial optimization problem that models sequential decision-making. In this setting, there are $n$ boxes, each containing a hidden value drawn from a known distribution. Opening a box incurs a known cost, and the value revealed is either $0$ or $\infty$. A scenario is said to be *covered* if at least one opened box reveals the value $0$ [FLT02]. The goal is to determine which boxes to open and which value to select, in order to minimize the total cost. This total cost includes both the costs of opening boxes and the value selected. The MIN-SUM SET COVER problem can be seen as a simplified version of the PANDORA'S BOX problem [Wei79]. There is a substantial body of literature focused on the MIN-SUM SET COVER problem, aiming to improve approximation guarantees and to investigate various generalizations. These generalizations encompass a wide range of settings, including but not limited to, selecting boxes without prior inspection, correlations among boxes, requirements to inspect boxes in a specific order, and situations involving more complex combinatorial constraints [MBMW05, AGY09, BGK10, AG11, SW11, ISVDZ14, INZ16, FKN+21, BBFT23]. Similarly, the PANDORA'S BOX problem itself is also well studied in the literature, with many works exploring its generalized settings to account for more complex information structures and decision-making frameworks [Dov18, FLX18, Sin18, GJSS19, BK19, BFLL20, CGT+20, SS21, BDP22, CGMT23, BC23, FLL23, GT23, BEFF23, MT23, DFH+23]. In addition, both MIN-SUM SET COVER and PANDORA'S BOX problems have been studied from a machine learning perspective, as it naturally combines algorithmic and learning aspects. Work in this direction has considered frameworks involving sample complexity and online learning models [EHLM19, GT22, BGKM20, FKK+20, CBCMP21, GHTZ21, BM23, BBT23, ACG+24, GKSW24].

# B  Solving Sudoku with frontier models

We evaluated the performance of several industrial-scale models from the OpenAI GPT [AAA+23], Google Gemini [TAB+23], and DeepSeek [GYZ+25] families, considering both general-purpose and reasoning-oriented versions. None of these models were able to produce correct solutions. The general-purpose models generate responses quickly but inaccurately, which made it feasible to test them on 1,000 Sudoku puzzles. In contrast, reasoning models require significantly more time for inference; therefore, we limited their evaluation to 10 representative samples. Additionally, we tested the models GPT-o1, GPT-o3, GPT-o4 mini, and Gemini-2.5 Pro. In every attempt, these models spent the entire inference process producing intermediate steps, without ever generating a final solution. For this reason, we have not included them in the result tables.

To further explore the impact of input representation on model performance, we tested three distinct formats for encoding Sudoku puzzles. Examples of each format are provided below.

- **Single-line format:** A single string of 81 digits (with 0 indicating empty cells), such as:
  000100000000309600070004023000060000004000806000120603050009290000
  380500800000
- **Grid format (9×9):** A standard 9-line grid with 9 digits per line, such as:
                      000100000
                      000030960
                      007000402
                      300006000
                      000040000
                      806000120
                      603050009
                      290000380
                      500800000
- **Triplet format (row, column, value):** A sequence of 3-digit numbers where the first digit is the row index, the second is the column index, and the third is the cell value, such as:

```
110 120 130 141 150 160 170 180 190 210 220 230 240 253 260 279 286
290 310 320 337 340 350 360 374 380 392 413 420 430 440 450 466 470
480 490 510 520 530 540 554 560 570 580 590 618 620 636 640 650 660
671 682 690 716 720 733 740 755 760 770 780 799 812 829 830 840 850
860 873 888 890 915 920 930 948 950 960 970 980 990
```

As shown in Tables 3–8, the evaluated models consistently fail to solve Sudoku puzzles across three different encoding formats. This failure occurs even when prompts explicitly suggest Sudoku solving strategies the models could follow. When prompts do not guide them to apply logical rules, the models tend to rely on external code or a backtracking algorithm. Backtracking relies on brute-force search rather than reasoning, making it less generalizable and frequently inefficient in challenging cases. Moreover, it can produce suboptimal or incorrect solutions, especially under time or resource constraints. As shown in the tables below, although Sudoku is a complex reasoning task, general-purpose models typically achieve, in most of the cases, slightly higher cell accuracy than reasoning-specific models, and they do so with significantly faster inference.

Reasoning models, tend to follow Sudoku-solving strategies more systematically than general-purpose models. However, they exhibit two key weaknesses. Firstly, their inference times are significantly higher. As shown in Table 4, the best-time performing general-purpose model completed a puzzle in 1.6 seconds on average, whereas the best reasoning model required 214.9 seconds in the single-line encoding format, an increase of 134.3%. Secondly, while reasoning models apply strategies in a more step-by-step fashion, they lack the flexibility to revise incorrect inferences. Once they make a mistake, they often continue along an invalid solution path, preventing them from arriving at a correct final answer. In some cases, they spend the entire inference process generating intermediate reasoning steps without ever completing the puzzle.

## B.1 Single-Line format

The prompt used in this case was as follows:

> **Prompt**
>
> ```
> Solve the following Sudoku puzzle using logical reasoning only.  Do
> not use or simulate any external code, algorithms, or automated
> solvers (e.g., backtracking).  Instead, apply human-style solving
> techniques such as lone singles, hidden singles, naked pairs,
> naked triples, locked candidates, pointing pairs, X-Wing, Swordfish,
> XY-Wing, unique rectangles, and other logical strategies typically
> used by experienced Sudoku solvers.  The puzzle is represented as
> a single line of 81 digits (0 means empty cell):  {puzzle_string}
> Return only the final solution as a single line of exactly 81 digits,
> with no extra explanation or formatting.
> ```

Table 3: Performance of two state-of-the-art LLMs on solving 1,000 Sudoku puzzles (in single-line format), evaluated by board-level accuracy and average inference time per puzzle.

| Model | Board Accuracy | Cell Accuracy | Time (sec./puzzle) |
|---|---|---|---|
| GPT–4o | 0.00% | 46.39% | 1.60 |
| Gemini–1.5 Pro | 0.00% | 34.9% | 3.20 |

Table 4: Performance of three reasoning state-of-the-art LLMs on solving 10 Sudoku puzzles (in single-line format), evaluated by board-level accuracy and average inference time per puzzle.

| Model | Board Accuracy | Cell Accuracy | Time (sec./puzzle) |
|---|---|---|---|
| GPT–o3 mini | 0.00% | 29.38% | 214.90 |
| Gemini–2.5 Flash | 0.00% | 22.22% | 253.80 |
| DeepSeek–R1 | 0.00% | 39.26% | 588.20 |

## B.2   9×9 format

The prompt used in this case was as follows:

```
Prompt

Solve the following Sudoku puzzle using logical reasoning only.  Do
not use or simulate any external code, algorithms, or automated
solvers (e.g., backtracking).  Instead, apply human-style solving
techniques such as lone singles, hidden singles, naked pairs,
naked triples, locked candidates, pointing pairs, X-Wing, Swordfish,
XY-Wing, unique rectangles, and other logical strategies typically
used by experienced Sudoku solvers.  The puzzle is represented as
9 lines of 9 digits each (0 means an empty cell):  {puzzle_string}
Return only the final solution as a single line of exactly 81 digits,
with no extra explanation or formatting.
```

Table 5: Performance of two state-of-the-art LLMs on solving 1,000 Sudoku puzzles (in 9×9 format), evaluated by board-level accuracy and average inference time per puzzle.

| Model | Board Accuracy | Cell Accuracy | Time (sec./puzzle) |
|---|---|---|---|
| GPT–4o | 0.00% | 44.62% | 1.60 |
| Gemini–1.5 Pro | 0.00% | 31.70% | 3.20 |

Table 6: Performance of three reasoning state-of-the-art LLMs on solving 10 Sudoku puzzles (in 9×9 format), evaluated by board-level accuracy and average inference time per puzzle.

| Model | Board Accuracy | Cell Accuracy | Time (sec./puzzle) |
|---|---|---|---|
| GPT–o3 mini | 0.00% | 30.99% | 265.70 |
| Gemini–2.5 Flash | 0.00% | 20.00% | 250.80 |
| DeepSeek–R1 | 0.00% | 41.48% | 510.40 |

### B.3 (Row, Column, Value) format

The prompt used in this case was as follows:

```
Prompt

Solve the following Sudoku puzzle using logical reasoning only.  Do
not use or simulate any external code, algorithms, or automated
solvers (e.g., backtracking).  Instead, apply human-style solving
techniques such as lone singles, hidden singles, naked pairs,
naked triples, locked candidates, pointing pairs, X-Wing, Swordfish,
XY-Wing, unique rectangles, and other logical strategies typically
used by experienced Sudoku solvers.  The puzzle is given as a list of
3-digit numbers separated by spaces.  Each number represents a cell:
the first digit is the row (1-9), the second is the column (1-9), and
the third is the value (0-9, with 0 meaning empty):  {puzzle_string}
Return only the final solution as a list of 3-digit numbers separated
by spaces, each indicating row-column-value (e.g., 123 for row 1,
column 2, value 3), with no extra explanation or formatting.
```

Table 7: Performance of two state-of-the-art LLMs on solving 1,000 Sudoku puzzles (in (Row, Column, Value) format), evaluated by board-level accuracy and average inference time per puzzle.

| Model | Board Accuracy | Cell Accuracy | Time (sec./puzzle) |
|---|---|---|---|
| GPT–4o | 0.00% | 27.54% | 6.00 |
| Gemini–1.5 Pro | 0.00% | 44.45% | 3.80 |

Table 8: Performance of three reasoning state-of-the-art LLMs on solving 10 Sudoku puzzles (in (Row, Column, Value) format), evaluated by board-level accuracy and average inference time per puzzle.

| Model | Board Accuracy | Cell Accuracy | Time (sec./puzzle) |
|---|---|---|---|
| GPT–o3 mini | 0.00% | 29.51% | 289.80 |
| Gemini–2.5 Flash | 0.00% | 25.19% | 245.40 |
| DeepSeek–R1 | 0.00% | 31.85% | 672.30 |

## C   Additional experimental details (Table 2)

Regarding the comparative results in Table 2, the RRN [PPW18] and Recurrent Transformer [YIL23] models were originally trained on the RRN dataset. We report their performance on the RRN test set based on the results published in their respective papers. To evaluate these models on additional test sets, we reproduced them using their publicly available code. We also retrained both models on our Random dataset by replacing the original training data with our own.

The Causal Transformer from [SDWP24] was trained on the filtered Kaggle dataset; we report its performance on both the Kaggle filtered and RRN test sets as published in their paper. The same holds for [KSK+25], where we report their results as published in their work. For the unfiltered Kaggle test set, [SDWP24] report the results of the Causal Transformer [SDWP24] in their paper, and we additionally computed the board accuracy on the full 3M dataset.

Regarding our approach, our Transformer-based model builds on Andrej Karpathy's open-source minGPT implementation (MIT License, GitHub Code).  All experiments were conducted using PyTorch [PGM+19] on a single NVIDIA A10G GPU.

We trained our model for 3M steps (about 168 GPU hours) with a batch size of 32 using the AdamW optimizer [KB15]. A linear learning rate schedule was applied: the rate was linearly increased to $1 \times 10^{-4}$ during the first 5 steps (warmup), then linearly decayed to zero over the remaining training steps.

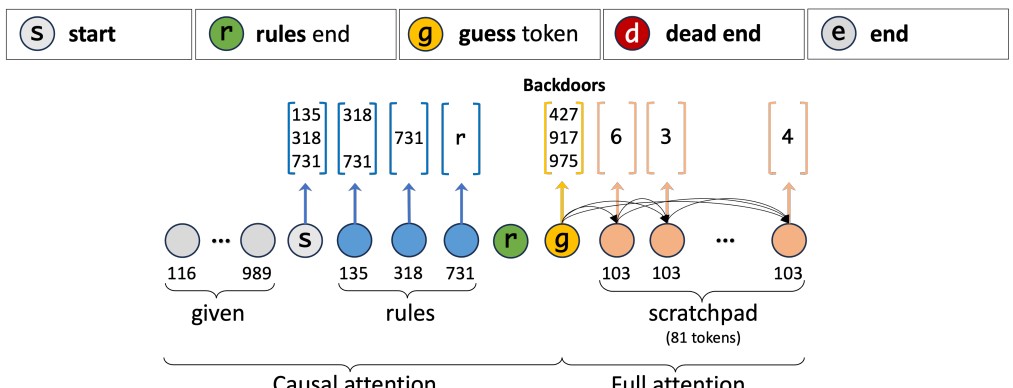

Figure 7: Expanded view of the guess node from Figure 5. The guess node is implemented as a guess token followed by a sequence of 81 thinking tokens (shown in orange), which form the scratchpad. Arrows represent full self-attention within the scratchpad and from the guess token to the thinking tokens. As in previous figures, all tokens attend causally to preceding tokens, but causal attention arrows are omitted for clarity.

## D  Additional details on optimizing guesses

Figure 7 illustrates our implementation of the guess mechanism. We design the guess node of Figure 5 as a guess token followed by a fixed sequence of 81 *thinking tokens*, which together form what we call the *scratchpad*. These thinking tokens correspond one-to-one with the 81 cells of a Sudoku board and are assigned fixed positional embedding indices from 1 to 81. Each thinking token is trained to predict the correct value of its corresponding cell (similarly to [YIL23]).

While the rest of the sequence follows standard causal attention, the scratchpad introduces a local window of full self-attention: the guess token attends to all 81 thinking tokens, and they attend to one another. Each scratchpad token also attends causally to all preceding tokens in the sequence. This structure is useful for the guessing step, as it allows the model to consider a full board solution and base its next guess on multiple coordinated cell predictions.

During evaluation, whenever the model outputs the guess token, we augment the input sequence by appending the scratchpad tokens, enabling a single forward pass with full attention between the guess token and the scratchpad. From this forward pass, we extract the model's output at the guess token and use it as the next generated token.

After this step, the scratchpad is removed from the input, and the generation proceeds normally in an autoregressive manner by appending the newly generated token to the existing sequence.

This approach maintains a next-token generation pipeline, while introducing a task-specific reasoning step at the guess token to leverage the Transformer's prediction power.

## E  Connection to the MIN-SUM SET COVER problem

We now describe the MIN-SUM SET COVER problem, in the terminology of our work. This models a depth-1 search problem where the goal is to find which guess would lead to a valid solution.

In this problem, there are $n$ potential choices one can take but only a subset $S$ of them are *valid*. While the exact subset $S$ is unknown, it is known that the subset is drawn from a given distribution $\mathcal{D}$. In the simplest variant, it costs 1 time unit to explore any choice and once a choice is made, one directly observes whether that choice was valid or not but receives no additional information. The goal in this problem is to find a policy $\pi$ that explores different options and minimizes the expected time until a valid solution is obtained. More formally, defining for every policy $\pi$ and every subset $S$, $T(\pi|S)$ the expected time until a valid choice $i \in S$ is made under policy $\pi$, we seek to find the best policy that minimizes:

$$\min_{\pi} \mathbb{E}_{S \sim \mathcal{D}} \left[ T(\pi|S) \right]$$

In the absence of additional feedback, the best policy takes the form of a permutation of all choices $1$ to $n$. For such a permutation $\pi$,

$$T(\pi|S) = \arg\min_{i=1}^{n} \pi_i \in S$$

Computing the optimal policy given a distribution over sets $\mathcal{D}$ is significantly challenging even if the distribution $\mathcal{D}$ is explicitly given. In fact, as [FLT02] show it is NP-hard to approximate the optimal policy better than a factor of $4$. Obtaining 4-approximate policy can be done in a number of different ways, e.g. via a greedy algorithm [FLT02] or solving a linear programming relaxation and performing randomized rounding [GT22].

**Theorem E.1** (From [FLT02]). *It is NP-hard to compute a policy that approximates the optimal by a factor better than 4. The greedy policy that always selects the most likely choice conditional on the set not being covered by the choices explored so far is 4-approximate.*

We note however, that these policies require heavy optimization and estimating the conditional distribution after several number of choices have already been explored. In contrast, naive methods that directly order choices by the marginal probabilities that the guess is valid are highly suboptimal and may end-up costing $\Theta(n)$-times more than the optimal policy.

**Example E.1.** *Suppose that $\mathcal{D}$ is simply the set $\{1, ..., n-1\}$ with probability $2/3$ and the set $\{n\}$ with probability $1/3$.*

*In such a case, a method based only on the marginals would explore all choices $1$ through $n-1$ before exploring the choice $n$ taking time at least $n/3$ in expectation. In contrast, the optimal policy is to first explore option $1$ and then option $n$ taking time $1 \cdot \frac{2}{3} + 2 \cdot \frac{1}{3} = \frac{4}{3}$ in expectation.*

Optimizing over MIN-SUM SET COVER policies is non-trivial and requires customized objectives. To simplify the optimization, we propose restricting focus to a simpler class of policies that are non-adaptive and do not depend on the choices made so far. Such policies are inherently randomized, and we treat $\pi$ as a probability distribution over the choices. Every time the policy $\pi$ selects a choice $i$ with probability $\pi_i$ irrespective of what has been tried so far.

**Definition E.1.** *A policy is called* non-adaptive *if the choice that it explores at every round is independent of the current time $t$ and choices explored up to time $t$.*

Evaluating the expected time until a valid choice is identified takes a simple form. At any time step, the probability that a valid choice is taken is $\pi(S) = \sum_{i \in S} \pi_i$. Therefore the expected time $T(\pi|S) = \frac{1}{\sum_{i \in S} \pi_i}$.

We thus seek to minimize

$$\mathbb{E}_{S \sim \mathcal{D}}\left[\frac{1}{\sum_{i \in S} \pi_i}\right] \tag{1}$$

This gives a natural loss function capturing the time to make a valid choice which is a convex optimization problem over the convex space of probability distributions over $n$ elements $\Delta(n) \triangleq \{\pi \in [0,1]^n : \sum_{i=1}^{n} \pi_i = 1\}$ and can thus be solved efficiently with stochastic gradient descent (SGD). To see this, note that the function $1/x$ is convex when $x > 0$, and thus $\frac{1}{\sum_{i \in S} \pi_i}$ is convex as a composition of a convex function with a linear function and the overall loss is convex as an expectation of convex functions.

Restricting over non-adaptive policies may come at a cost as some choices may be explored multiple times increasing the time until a solution is found. However, the following theorem that we establish guarantees that the total time of the optimal non-adaptive policy is at most $O(\log n)$ times that of the optimal permutation.

**Theorem E.2.** *For any distribution $\mathcal{D}$ over sets $S \subseteq [n]$, it holds that for any permutation $\tau$:*

$$\min_{\pi \in \Delta(n)} \mathbb{E}_{S \sim \mathcal{D}}\left[\frac{1}{\sum_{i \in S} \pi_i}\right] \leq H_n \cdot \mathbb{E}_{S \sim \mathcal{D}}\left[\arg\min_{i=1}^{n}\{\tau_i \in S\}\right]$$

*where $H_n = 1 + \frac{1}{2} + \frac{1}{3} + \cdots + \frac{1}{n} = \Theta(\log n)$ is the $n$-th harmonic number.*

*Proof.* Fix a permutation $\tau$ and let $q_i$ be the probability that a random $S \sim \mathcal{D}$ is covered in exactly $i$ steps. The cost of permutation $\tau$ i.e. $\mathbb{E}_{S \sim \mathcal{D}} [\arg\min_{i=1}^{n} \{\tau_i \in S\}]$ is given by $\sum_{i=1}^{n} q_i \cdot i$.

Without loss of generality assume that $\tau_i = i$. Now consider a non-adaptive policy $\pi$ that sets the probability $\pi_i \propto \sqrt{q_i}$ i.e. $\pi_i = \frac{\sqrt{q_i}}{\sum_{j=1}^{n} \sqrt{q_j}}$.

We now have that

$$\mathbb{E}_{S \sim \mathcal{D}} \left[ \frac{1}{\sum_{i \in S} \pi_i} \right] \leq \sum_{i=1}^{n} q_i \frac{1}{\pi_i} = \left( \sum_{i=1}^{n} \sqrt{q_i} \right)^2$$

where the inequality follows from the fact that the $q_i$-fraction of sets $S$ that are covered in $i$ steps according to $\tau$ must contain the element $i$.

To complete the proof we need to show that $\left( \sum_{i=1}^{n} \sqrt{q_i} \right)^2 \leq (\sum_{i=1}^{n} \frac{1}{i})(\sum_{i=1}^{n} i q_i)$ which follows from the Cauchy-Schwartz inequality on the sequences $a_i = \sqrt{i q_i}$ and $b_i = \frac{1}{\sqrt{i}}$. $\qquad \square$

**Remark E.1.** *Using the loss function of Equation* (1)*, results in solutions with bounded approximation to the optimal policy. In contrast, if one treats the problem as a multi-class classification task and attempts to use a loss function such as a weighted Cross-Entropy Loss*

$$\mathbb{E}_{S \sim \mathcal{D}} \left[ \alpha_S \sum_{i \in S} \log \frac{1}{\pi_i} \right]$$

*the approximation would be much worse. This is because the Cross-Entropy loss can be rewritten entirely as a function depending only on the marginals $\sum_i \beta_i \log \frac{1}{\pi_i}$ for some given weights $\beta_i$ and as we established in Example E.1, this leads to paying a linear multiplier in the total expected time.*

### E.1 From Sudoku to contextual MIN-SUM SET COVER problem

Mapping: the choices are the moves to be suggested as backdoors, and the goal is to efficiently figure out which backdoor given as a hint would result in a correct solution.

While the above analysis considers a fixed distribution over sets, in our setting we wish to exploit information about the instance to quickly pin-down the correct choice. While in our setting there is no stochasticity as all information about which choices would lead to solutions is directly encoded in the initial state, such a mapping may be very incompressible and may not be representable even by large NNs. As such stochasticity arises due to this loss of information. We can thus view the problem as a generalization of MIN-SUM SET COVER where the current state is given as a context $\vec{x}$ and the goal is to learn a mapping from context to a search policy (distribution over choices). In this generalization, we are given a distribution $\mathcal{D}$ over contexts $\vec{x} \in \mathbb{R}^d$ and sets $S \subseteq [n]$, a class $\Pi$ of contextual policies $\pi : \mathbb{R}^d \to \Delta(n)$ and we seek to minimize:

$$\min_{\pi \in \Pi} \mathbb{E}_{(\vec{x}, S) \sim \mathcal{D}} [T(\pi(\vec{x}) | S)]$$

If the class $\Pi$ contained all linear mappings i.e. $\pi = Ax$, the loss function presented in Equation (1) would remain convex as a function of the entries of $A$, and one can again optimize it via SGD. In general, we apply our method over mappings given by complex NNs and Transformers so optimality is not-guaranteed but hopefully we can converge to good local minima.

What is guaranteed however, is that as long as every choice gets assigned non-trivial probability of being selected we will eventually arrive at a solution.

### E.2 Multiple-levels of guessing

Our discussion so far focused on single level of guessing which corresponded to MIN-SUM SET COVER. One can define a generalization of the problem on arbitrary trees. We can still consider randomized non-adaptive policies as performing random walks in the tree until a node that corresponds to a valid solution is reached. Optimizing over non-adaptive policies would boil down to estimating the transition probabilities of a (absorbing) Markov Chain so that the time to solution (time to absorption) is minimized.

While this is a very interesting extension to explore both theoretically and experimentally, this comes with additional challenges that were out of scope of the current work. We plan to do explore them however in future work.

# F   Libraries of data generators

## F.1   SudokuPy: a fast Sudoku puzzle generator

In this section, we introduce SudokuPy; a fast Python library, primarily implemented in C, that generates uniformly random Sudoku puzzles in a principled way from the full distribution of valid Sudoku grids. For combinatorial problems like Sudoku, it is essential to generate diverse, truly random instances rather than rely on static datasets, allowing stream-like training and avoiding overfitting issues as every puzzle is only seen once. SudokuPy addresses this need by enabling on-the-fly generation of puzzles along with their corresponding solutions, making it particularly suitable for large-scale data generation and advanced AI training for standardized benchmarks in research. The library ensures principled and efficient puzzle generation by sampling from the full distribution of valid grids using precomputed permutation tables and optimized indexing. It includes a high-performance C-based solver to validate and generate puzzles, significantly improving computational speed. SudokuPy can be installed as the Python package sudokupy. It is distributed under the MIT License, ensuring free accessibility, and the distribution can also be reproduced following the description below.

The core functionality of our generator is described and illustrated in Figure 8. In this figure, the input to our library function is shown in blue: the number P, which specifies how many Sudoku puzzles to generate. The output is highlighted in green and labeled S; it represents the generated Sudoku puzzles.

The process starts by generating a uniformly random number between 1 and 6,670,903,752,021,072,936,960, which corresponds to the total number of valid Sudoku puzzles. This number is then mapped to a fully solved Sudoku board using precomputed permutation tables and the functions board_encode and board_decode, which together form a one-to-one mapping. In the next stage, the generator gradually removes numbers from the completed board using the puzzle_generator function. After each removal, the generator verifies that the puzzle still has a unique solution by calling the puzzle_solver function. While puzzle_generator handles the random deletion of entries, puzzle_solver ensures that the puzzle remains valid. The puzzle_solver is C-based function integrated with our package, ensuring high performance even on systems lacking a native C compiler. For user convenience, we provide the black-box function named as sudokupy_generator, which takes an integer as input and returns that many random Sudoku puzzles along with their corresponding solutions.

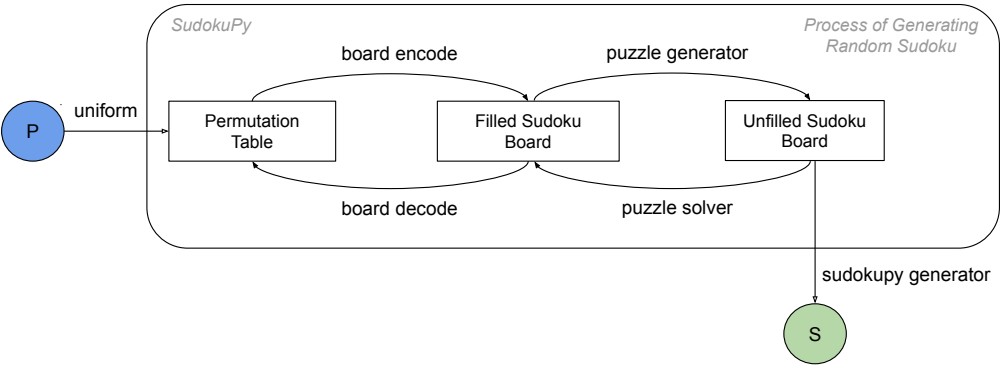

Figure 8: Main process and features of the uniform Sudoku puzzle generator, SudokuPy

### F.2 Transcripts library

The Transcripts library is primarily implemented in C, and it can be installed and used as a Python package. It provides transcripts for a wide range of problems, including Sudoku and 1-in-3 SAT problem which are used in this work. It automatically supports any combinatorial problem as long as the user can implement the following functions.

```c
// Generate a (random) problem instance
Problem generate_instance(GeneratorParameters *params);

// Given a problem instance p encodes it into tokens
int problem_tokens(Problem *p, int *tokens);

// Provides the initial partial solution to the problem p.
   Potentially empty.
PartialSolution initial_solution(Problem *p);

// The tokens corresponding to valid guess actions
int guess_next_tokens(PartialSolution *s, Problem *p, int *tokens);

// If a guess fails, provide the alternative tokens to try next.
int alternatives_next_tokens(int guess_token, PartialSolution *s,
   Problem *p, int *tokens);

// Find the next action tokens according to logic rules
int logic_next_tokens(PartialSolution *s, Problem *p, int *tokens);

// Given a partial solution obtains a new partial solution by
   applying the given action token
PartialSolution apply_token(PartialSolution *s, Problem *p, int
   token);

// Checks if the solution is complete or whether errors exist
int is_complete(PartialSolution *s, Problem *p);
```

The transcripts library uses this interface to run a DFS search on top of the partial solutions to the problem and provides complete transcripts along with multi-targets for the corresponding combinatorial problem.

## G  1-in-3 SAT problem

**Definition.**  Given $N$ Boolean variables $x_1, \ldots, x_N$ and a Boolean formula $\phi$ with $M$ clauses in 3-Conjunctive Normal Form

$$\phi = C_1 \wedge \cdots \wedge C_M, \quad \text{where } C_i = (\ell_{i1} \vee \ell_{i2} \vee \ell_{i3}),$$

and each literal $\ell_{ij} \in \{x_k, \neg x_k\}$, the *1-in-3 SAT* problem asks for a truth assignment to the $N$ variables such that *exactly one literal in each clause is true*.

**Instance generation.**  We use the following concrete process for generating 1-in-3 SAT instances that is popular in the literature. Starting from a random assignment for the $n$ variables, we create a random clause using 3 random literals on 3 distinct variables and reject if the clause is not satisfied by exactly 1 literal. We repeat this process until $m$ clauses are selected. In contrast to the Sudoku task, this is not guaranteed to have a unique solution but as the initial chosen assignment is a solution it is guaranteed to be solvable.

**Transcript generation.**  To generate the transcript, we make guesses by assigning a value to a random variable. We also apply the following logic rules to figure out additional assignments if possible:

- **Positive inference.** If a clause has a literal that was fixed to be positive, the remaining two literals must be negative.

- **Negative inference.** If a clause has a literal that was fixed to be negative, the other two literals must have different values. While this does not give us a new value, it allows replacing, instances of one literal with the negation of the other. Combined with the following rule this lets us infer additional assignments.
- **Same variable inference.** If a clause contains a variable $x$ twice, the value of the other literal is automatically determined. If the variable appears with opposite signs, the other literal must be negative while if it appears with the same sign, the other literal must be positive and the literals corresponding to the variable $x$ must be negative.

We generate the transcript by applying the above rules whenever possible and make a guess if no additional assignment can be inferred.

Example of a 1-in-3 SAT transcript with $N = 25$ and $M = 15$ is provided below.

```
(14, -13, 24), (18, -5, -12), (7, -10, -4), (15, 5, -11),
(-3, -21, -25), (25, -13, -11), (-17, -7, -8), (12, 1, -16),
(-22, 18, -24), (-8, 19, -17), (-6, 8, -24), (-3, 2, -1),
(21, -22, -15), (8, 14, -4), (23, -22, -20), s, r,
L1, 4, r,
L2, 18, 12, -1, 5, 22, 11, 3, -15, -2, 16, -21, 24, 13, 25, -14, 6, 8, r,
L3, -17, -19, 7, 10, r,
L4, -9, r,
L5, 20, r,
L6, 23, r, e
```

**Metric.** We evaluate the performance in terms of board accuracy, as in Sudoku. In this case, a board is correct if all the clauses have exactly one satisfied literal.

**Results.** In the 1-in-3 SAT problem for $N = 25$ and $M = 15$ we achieve 99.10% board accuracy. During evaluation, for each variable we keep the last assigned value in the generated sequence.

