# OpenReview forum: "Teaching Transformers to Solve Combinatorial Problems through Efficient Trial & Error"
_NeurIPS.cc/2025/Conference — NeurIPS 2025 poster_

### Official Review · Reviewer_adCH · 2025-06-07

**Clarity:** 3
**Significance:** 2
**Originality:** 3
**Rating:** 5
**Confidence:** 4

**Summary:**

This paper studies transformer models for combinatorial problems, with a specific focus on NP problems. This has been studied a lot in recent years by the community, as covered in the related works section of the paper. The authors obtain gains of >10% for full solution accuracy using architectures from prior work with their new data generating library, which also allows for test set generation.

The authors then make three key addtions to obtain another gain of >6% to obtain an accuracy of 99%. Firstly, a method for representing the solving process as a series of discrete actions, secondly allowing the model to learn backtracking through curated training data and finally using a novel loss function which is not one hot to allow for probability of the model to be placed on all correct possible next moves. The change is loss function is backed by both theoretical and empirical results. One empirical validation of this new loss function is in Figure 6 where the new loss function allows more than 80% of puzzles to be solved in fewer guesses than an oracle.

**Questions:**

1. Is the Deepseek-R1 “Time” in number in table 1 large as it was run locally and all others are run on APIs? Either way, it would be very useful to have exact API references for all models in Table 1.

2. Where would the r,c,v + multiple targets line be on Figure 3 left?

3. Please see weakness 1, on "additional time/memory taken to train and inference using the new method."

Discussion/progress on weakness 1 is most likely to impact my score.

**Ethical Concerns:**

["NO or VERY MINOR ethics concerns only"]

**Final Justification:**

## Why not higher
- I am not sure the paper will have "groundbreaking impact" as a 6 requires

## Why not lower
- I think this paper will have impact on the sub area of AI that sudies what algorithms transformers can learn
- The evaluation scheme is rigorous and compares multiple models
- The authors will open source code for reproducibility
- No unaddressed ethical considerations

**Limitations:**

I would like to see a little more on the limitations of this work compared to specialised/classical SAT solvers.

**Quality:**

3

**Strengths And Weaknesses:**

# Strengths
- Strong related works section which places the paper well in context.
- The introduction and study of a novel loss function for combinatorial problems and new method to train models to learn to backtrack.
- The promise to open source a new library to make it easier for the community to study NP problems in the future.

# Weaknesses

- There is a lack of discussion on the additional time/memory taken to train and inference using the new method. The authors compare to non-backtracking techniques which should use less tokens/compute to generate their solutions, although these come with the down sides the authors highlight throughout their paper there are other ways to compare to these methods on a FLOPs/output-tokens comparable axis.
	- For example, if the new backtracking technique takes twice the compute/tokens then we could run the non-backtracking approaches twice with a non-zero temperature to see if they generate the correct solution either time. One thing to note is that as the multiple attempts can be completed in the batch dimension the exact comparison here is intricate but inference FLOPs or total tokens output could be good options to make comparisons fair.
	- Additionally, in Table 1 the authors analyse at reasoning models for Sudoku. A reasonable baseline could be do GRPO directly on sudoku problems, as I believe “Aha movement” in GRPO (https://arxiv.org/pdf/2501.12948) does allow for backtracking.

- It would be nice to contextualise in the conclusions outside of the language modelling community. For example, in the context of how fast traditional SAT solvers are, and highlight the future directions this work allows, highlighting the connection to all NP-Complete problems more directly for non topic specific readers.

---

> ### Author Rebuttal · Authors · 2025-07-31
>
> We thank the Reviewer's for their time and thoughtful feedback, which has provided valuable guidance for refining our paper. We have carefully considered each comment and have revised the manuscript accordingly. Our detailed responses to each point are provided below.
> 1) Thank you for your thoughtful comments regarding the time, memory, and FLOPs considerations.
>
>     Regarding the input/output tokens and compute: in our work, we use the same vanilla decoder-only Transformer architecture as in [SDWP24] (see line 143 and Appendix D). This choice ensures that the underlying model capacity, token processing cost, and FLOPs per step remain comparable.
>
>     The purpose of Table 1 is mainly to highlight that even very large, high-capacity industrial language models (and even with their reasoning mode enabled) spend significant resources attempting to solve Sudoku, yet ultimately fail to solve the puzzles correctly. This observation arises the need for effective, alternative reasoning strategies in combinatorial problems.
>
>     As stated in the introduction (and now made more explicit in the revised draft), our primary goal was to demonstrate that vanilla Transformers can acquire a generalizable and interpretable reasoning strategy for combinatorial problems such as Sudoku, rather than to optimize absolute compute efficiency or minimize FLOPs.
>
>     It might be possible to create transcripts, for example by using external Sudoku solvers as in [SDWP24], that can produce a shorter solution path. However, this requires sophisticated, predefined solving algorithms/strategies to generate training data, which is not feasible for all Sudoku puzzles. One of the key advantages of our approach is that it is based only on the fundamental Sudoku rules and simple clever guessing, without relying on handcrafted strategies.
>
> 2) We added additional relevant references and discussion about SAT solvers.
>
>     Please note that although classical SAT solvers can indeed solve Sudoku puzzles very quickly, they are not learned systems. Our goal is not to outperform SAT solvers, but rather to show that vanilla Transformers can acquire a generalizable and interpretable reasoning strategy. This is why a direct comparison with SAT solvers is not appropriate, and instead we chose state-of-the-art works like [SDWP24] as our main baseline (see Table 2).
>
>     In particular, we achieved state-of-the-art results already with our first approach (imitation learning), and then improved further with our second approach (beyond imitation learning), again reaching state-of-the-art performance.
>
> 3) We ran DeepSeek-R1 also via its API, for which we paid usage fees. Further details regarding the prompts used are provided in the Appendix to ensure clarity.
>
> 4) Our intuition is that the r, c, v + multiple targets line would lie above the blue curve but below the orange and green curves in Figure 3.
>
> We hope these explanations make our framework and decisions clearer. Please let us know if there is anything else you would like us to elaborate on.

---

> > ### Comment · Reviewer_adCH · 2025-08-01
> > **Rebuttal Response**
> >
> > Thank you for your thoughtful reply. I think my weakness 1 is still not quite fully addressed.
> >
> > I acknowledge table 1 is a demonstration that this problem is hard as frontier models cannot solve it. My concern is the new backtracking approach requires more output tokens. Hence, even if the same architecture is used as the vanilla baseline, more compute is required. Maybe you have the number of tokens output from each approach to hand to clarify how much the backtracking costs? Or some other similar measure?

---

> ### Author Response · Authors · 2025-08-02
>
> Thank you for your follow-up question Reviewer adCH!
>
> In terms of output tokens for the Sudoku task, our method (causal transformer + backtracking) generates on average ~82 tokens until reaching a solution.
>
> For comparison, the optimal (or minimum) number of output tokens for our method would be 81−ngiven+2 tokens. The '+2' accounts for the <start> and <end> tokens, and ngiven is the number of initially filled cells, which is typically in the range of 18 to 34. This optimal case occurs when the model correctly fills all the remaining cells, one token at a time. In contrast, the optimal length for the baseline causal transformer [SDWP24], which doesn't use backtracking, is 3*(81−ngiven). The factor of 3 is because they use three tokens for each move.
>
> This shows that, on average during evaluation, our method outputs fewer tokens than even the optimal output length for the baseline method [SDWP24]. A further advantage is that because their initial input is larger, each token generation is slower and more computationally heavy for their approach.
>
> You are correct that when backtracking is involved, the output length could become quite large. However, our key idea is to iteratively combine logic rules with backtracking, which significantly reduces the search space and therefore the computational overhead of backtracking. The degree to which the search space is reduced depends on the number and effectiveness of the logic rules used. In the Sudoku case, the very simple rules that define the task were enough to accelerate our method, making it both viable and efficient despite the use of backtracking. Similar efficiency was observed in the 1-in-3 SAT task with the rules employed there.

---

> > ### Comment · Reviewer_adCH · 2025-08-02
> > **Rebuttal Response 2**
> >
> > Thank you for the clarifications, I think I slightly misunderstood but now see (maybe this could be detailed a little more in the paper). I will recommend acceptance for this paper.

---

> > > ### Author Response · Authors · 2025-08-02
> > >
> > > Thank you very much for the insightful discussion and for your intention to recommend accepting our paper. We will definitely incorporate the points we discussed above into the revised draft to make everything clearer.

---

### Official Review · Reviewer_ekMH · 2025-06-24

**Clarity:** 2
**Significance:** 2
**Originality:** 2
**Rating:** 3
**Confidence:** 4

**Summary:**

The paper presents a framework for solving combinatorial problems such as Sudoku using a standard decoder-only Transformer model. The key innovation lies in combining supervised imitation learning of basic Sudoku rules with a trial-and-error Depth-First Search (DFS) strategy. This enables the model to make informed guesses, backtrack when encountering conflicts, and eventually solve problems that cannot be addressed by heuristic rule application alone. Additionally, the paper introduces a novel loss function inspired by the Min-Sum Set Cover problem to optimize inference efficiency. The proposed method achieves near-perfect accuracy (99%) on Sudoku and 1-in-3 SAT tasks, outperforming prior neural and neuro-symbolic baselines.

**Questions:**

- What exactly is supervised imitation learning in your setup? A concise explanation would be helpful for readers unfamiliar with this technique.
- What optimiser, GPU type, and training time (in GPU hours) were used? Reproducibility would benefit from detailed training settings.
- The algorithm starting from the line 214 seems to be the main contribution. While the generation of training transcripts is explained, how exactly is this used during training and inference? What is the architecture of the network that takes these transcripts and learns from them?
- How does the algorithm described around line 214 differ from that of the Causal Transformer? Based on your explanation, the main differences seem to be: (1) using 4 inference rules instead of 7, (2) using a more compact encoding by removing commas between row, column, and value triplets, and (3) adopting a multiple next-token prediction loss. Are there any other differences? Given that the Causal Transformer is a key baseline, the paper should explain this comparison more clearly.
- In Table 2, why is there no test result for Causal Transformer on the Random dataset? Is it infeasible to evaluate?
- In Table 2, is the Causal Transformer trained on the Kaggle filtered dataset only? It would be important to determine whether the performance gain of your model is due to the dataset change during training or the proposed methods in Section 2. An ablation study is needed to identify which factors contributed most to the performance improvement.
- What do the terms “lower bound” and “upper bound” in Figure 6 represent? Is the lower bound equivalent to the “ours” model described in Section 2? The upper bound seems to assume access to the correct values of the remaining cells, functioning as an oracle. In that case, what does “guess” mean when the correct values are already known? Does it refer to steps where none of the four Sudoku rules can definitively determine the next number?
- The paper’s approach to forward guessing and backtracking in Sudoku closely resembles the DPLL algorithm used in SAT solving. In DPLL, deterministic rules are applied until no further progress is possible, then a guess is made; if a contradiction arises, the process backtracks. A notable extension is CDCL (Conflict-Driven Clause Learning), which learns from conflicts to avoid similar mistakes. Adopting such strategies could help extend this method to more general NP-class problems.

**Ethical Concerns:**

["NO or VERY MINOR ethics concerns only"]

**Final Justification:**

I still have some doubts about the applicability of this methodology to general combinatorial problems. While the authors demonstrate high accuracy on the 1-in-3 SAT problem in the appendix, I feel that the discussion in the main text is insufficient to fully support this claim. Specifically, the main text should have included what kind of 1-in-3 SAT instances were used in the experiments, whether they are representative or meaningful, how difficult they are, or whether they were derived from any practically challenging combinatorial problems. Additionally, the proposed contribution seems to be an incremental improvement based on Causal Transformer, so it would be helpful if the authors could better clarify what makes this work sufficiently novel to warrant acceptance.

**Limitations:**

This paper presents a framework that combines symbolic reasoning with Transformer-based learning in a novel way. Its success on Sudoku shows promise. However, the paper would benefit greatly from improved clarity in technical exposition, more extensive discussion of generalisability to broader NP tasks and general-purpose natural language processing, and inclusion of relevant related work in SAT-solving and symbolic learning in LLMs. If these areas are addressed, this work is likely to make a meaningful contribution to the field of combinatorial reasoning with language models.

**Paper Formatting Concerns:**

There is no formatting issues in this paper.

**Quality:**

2

**Strengths And Weaknesses:**

Strengths
---
- The model leverages a pure Transformer architecture (GPT-2 variant) without requiring architectural modifications or external solvers.
- The model achieves 99% accuracy on standard Sudoku, demonstrating strong reasoning ability.
- The authors provide a custom C-based puzzle generator and training transcript generator to support large-scale evaluation and reproducibility.
- The authors demonstrate fair comparisons across multiple datasets and training regimes (Kaggle, RRN, Random) in their experiments.

Weaknesses
---
- **Length and Structure of Introduction**: The introduction is overly long and merges motivation, background, and methodology. This impairs clarity about the specific contributions. Reorganising the content (e.g., by moving methodology details to a separate section) would improve readability.
- **Lack of Training/Inference Time Reporting**: While the motivation includes time-per-puzzle as a crucial factor (Table 1), the paper does not report actual training or inference times, which weakens claims about efficiency.
- **Insufficient Detail on Generalisation to NP Problems**: While the method is said to be applicable to general NP tasks, concrete explanations on how domain-specific design choices (e.g., tokenisation 111--999, rule application mechanics, backdoors) translate to other problems are not enough. The treatment of the 1-in-3 SAT task is not found in the main text.
- **Neural Network Specification**: The paper does not clearly explain what inputs the Transformer receives and what outputs it produces. For example, while the transcription format and guess strategy are described, the Transformer input/output format and parameterisation are not fully formalised.
- **Theoretical Contributions Need Clarification**: In Section 3, the comparison between two loss functions (L1 and L2) is said to be grounded in theory, but the intuitive significance of the findings is underexplained. More discussion of what the results mean practically would help.
- **LLM Framing and Generalisation**: It remains unclear how this model fits into the LLM paradigm. Unlike general-purpose LLMs that process natural language prompts, this model uses highly structured tokens. The authors should clarify whether the model is applicable to natural language-stated logic puzzles or if its scope is limited to symbolic encodings.
- **Missing Related Work on SAT Solvers**: The related work focuses heavily on Sudoku solvers. More references with neural SAT solvers (e.g., NeuroSAT, GNN-based approaches) would help understand the work within the broader literature.

---

> ### Author Rebuttal · Authors · 2025-07-31
>
> We sincerely thank the Reviewer for their constructive feedback and valuable suggestions. Their comments have helped us identify areas where additional clarification and improvements were needed. Below, we address each of their points in detail:
>
> 1) Our framework applies to any problem in NP and while we only illustrate this for Sudoku and 1-in-3 SAT, it is really easy to integrate any other NP problem. Note that since 1-in-3 SAT is NP-complete, by the Cook-Levin theorem, any problem in NP can be written as 1-in-3 SAT.
>
>     However, some problems may benefit from problem-specific optimizations and therefore, we provide a library in Appendix F2 that lets one generate transcripts for any problem in NP by providing the appropriate components.
>
>     Our framework has essentially 3 components that are learnable by a transformer. Solution generation, Solution verification and Guess generation (Hints).
>
>     For any problem in NP, these components are simple to obtain. Problems in NP have short certificates meaning that few guesses are needed to arrive at a solution. Moreover, problems in NP have efficient solution verification meaning that it is easy to verify correct solutions. The challenging part for NP (or NP-complete) problems is to arrive at a good solution and it is an open problem whether this can be done efficiently. As we discussed in our previous response, our goal is to optimize efficiency by automatically learning correct and useful guesses.
>
>     While our exploration was limited to problems in NP, our framework can also apply in other domains, as Reviewer stHu mentioned, with efficient verification like verification of mathematical proofs or program correctness. In fact a similar pipeline was used in the recent work https://arxiv.org/abs/2507.15855 for solving math problems for IMO 2025
>     where hints like "mathematical induction" and "analytic geometry" where given to a solver that produced proofs that were subsequently verified using an LLM-based proof verifier.
>
> 2) Imitation learning refers to training a Transformer to predict the next step in rule-based reasoning transcripts generated by a symbolic solver, or handcrafted solvers such as the [SDWP24] approach.
>
>     In our setup, imitation learning means that the model is trained to follow the sequence of logic steps depicted in the transcripts, i.e. iteratively apply the simple rules and perform guesses in a DFS way. The results of this approach were state-of-the-art compared to the [SDWP24] method, and our framework is simpler since it does not rely on external solvers, unlike [SDWP24].
>
> 3) Thank you for your comments regarding the distinction between our work and the previous Causal Transformer [SDWP24].
>
>     There seems to be a misunderstanding in your comment (1). The [SDWP24] approach creates data transcripts based on 7 handcrafted Sudoku-solving strategies (not just rules). This means that good solving algorithms must first be predefined to generate training data. In contrast, our approach relies solely on the fundamental rules of Sudoku combined with simple guessing, without requiring handcrafted strategies.
>
>     In our imitation learning setup, we do not allow the model to make wrong guesses and backtrack. In the beyond imitation learning setup, however, wrong guesses and subsequent corrections are permitted, enabling a more flexible reasoning process. This extension achieved further state-of-the-art results, outperforming our earlier imitation learning approach.
>
>     We also kept our model architecture identical to [SDWP24] (see line 143 and Appendix D) to ensure a fair comparison. Your points (2) and (3) are correct and represent additional differences between the two works.
>
>     Another important difference is that, we formulate the guessing process in the context of the Min-Sum Set Cover problem. We introduce a novel loss, which is described and theoretically justified in Appendix B.
>
> 4) In our work we indeed focused on training Transformers for combinatorial problems with problem-specific tokens rather than free-form text. This type of a more succinct/structured representation allowed for efficiency in training and evaluation.
>
>     However, our pipeline is based only on decoder-only Transformers (like LLMs) and can be directly combined with natural language expressing the transcripts in plain English. e.g. instead of using token 134 we could instead write "I place number 4 in row 1 and column 3". Also we could be more descriptive in some tokens, e.g. when a deadend token appears, we could describe in natural language 'conflict occurred; no valid value can be placed in row X and column Y." However in this case training and evaluation would be far less efficient as the input sequence length increases. We might also need a larger Transformer model to capture the increased complexity of this expanded vocabulary.
>
>     So in principle, our method is not limited to input sequences of symbolic encodings. It could be applied to any combinatorial puzzle, as long as we can apply some logic rules to the problem and we can express them in natural language in a consistent way.
>
> 5) The training data transcripts for the Transformer consist of Sudoku puzzles represented in RCV format (row–column–value). This defines the sequence structure and includes multiple target predictions and guesses. The training input is exactly as illustrated in Figures 2 and 5.
>
> 6) Sorry for the confusion regarding the oracle terminology and we will clarify further in the paper to make it clear.
>
>     We provide two theoretical oracles as a reference to help parse the results, one upper-bound and one lower-bound, and are unrelated to transformers or specific architectures.
>
>     The upper-bound oracle corresponds to knowing the full Sudoku solution but not which cells are backdoors. This corresponds to guessing the correct value for a random unfilled cell (after the basic rules are applied) but this guess may not be a backdoor resulting in a restart.
>
>     The lower-bound is similar but does not assume we know the solution.  After the basic rules are applied, it chooses a uniformly random (cell, value) combination that does not conflict with the existing values, i.e. appears valid but may not be correct and restarts if the combination is not a backdoor. Note that a backdoor must necessarily be correct as the solution to a Sudoku is always unique.
>
>     Regarding the choice of L1 and L2 losses, please note that the L2 loss corresponds to the standard Cross-Entropy loss, while the L1 loss is our novel contribution, derived from the Min-Sum Set Cover problem as discussed in Section 3 and Appendix B. You can interpret this L1 loss in terms of a geometric distribution: we treat each good guess as a Bernoulli trial where a “success” means finding a valid backdoor. Not every correct guess actually works as a backdoor, so each attempt can succeed or fail. The goal is to minimize the expected number of trials needed to find a working backdoor; analogous to minimizing the expectation in the geometric distribution.
>
>
> 7) What Table 1 mainly aims to show is that even these large, high-capacity industrial models spend a significant amount of time attempting to solve Sudoku, yet in the end, they fail to actually solve anything.
>
>     In our case, the inference time is in the order of milliseconds (while achieving ~99\% accuracy).
>
>     Regarding training, we trained our model for 3 million steps with a batch size of 32 using the AdamW optimizer, on a single NVIDIA A10G GPU. We omitted detailed training and inference times because they depend heavily on the specific computational resources used. In our case (single A10G GPU), the training time took approximately 168 GPU hours.
>
> 8) Thanks for mentioning the SAT solvers related work and DPLL algorithm. We have added relevant references and discussion.
>
> 9) Regarding Table 2, the [SDWP24] model is trained and tested only on Sudoku puzzles that can be solved using the 7 handcrafted strategies they define. Randomly generated Sudoku puzzles cannot always be solved with these strategies, so it is not even possible to create full training transcripts for all such data.
>
>     For this reason, we provide an approximate accuracy for [SDWP24] on random puzzles (thus marked with * in Table 2). The results for [SDWP24] on the Kaggle filtered and RNN datasets are taken directly from their paper.
>
>     The difference in performance arises from our general framework, as described earlier, and not from the choice of dataset. Note that we compare our model against several state-of-the-art approaches, and both of our methods outperform them (see Table 2).
>
> We hope these explanations make our approach and choices clearer. Please do not hesitate to ask if you have additional questions; we would be glad to provide further details.

---

> ### Comment · Reviewer_ekMH · 2025-08-05
>
> I appreciate the authors for their clarifications via rebuttal. While the rebuttal adds some useful context, I believe the paper still needs improvements in clarity and exposition before it can be considered for acceptance.

---

> > ### Author Response · Authors · 2025-08-06
> >
> > We are glad that our rebuttal helped clarify our work. In the revised draft, we have considered all the Reviewers' comments to further improve the clarity of our paper.

---

### Official Review · Reviewer_stHu · 2025-07-02

**Clarity:** 4
**Significance:** 3
**Originality:** 3
**Rating:** 5
**Confidence:** 3

**Summary:**

A new strategy to tackle NP problems using transformers is proposed (exemplified by the classic Sudoku puzzle) - without external tools or code. The training and operation of the model is in 2 phases: learning basic Sudoku rules via imitation learning, and learning to guess cell values efficiently via a depth-first search - eventually getting to single-guess solution flows. A generator for Sudoku puzzles in a uniform distribution - for training and testing - is developed and published.

The new method is applied to several test datasets - achieving SOTA results.

**Questions:**

Line 141: The oracle is the same system that generated solutions for the imitation training stage? If so, I’m not sure I understand how we know that its solutions are optimal - or how close to optimal they are? It’s still a strong result if the transformer outperforms the training data substantially - just not clear how to attribute it specifically to the “backdoor guess” step.

Figure 3 - if the r,c,v representation counts each step as 3 tokens, so the number of actual steps seen by the model is 3 times lower - yet achieves similar (even if slightly lower) results? I probably misunderstood something.

Line 293: Is the “backdoor property” a known feature of Sudoku? Is it an empirical observation based on your generation of training data?

Have you tried experimenting with a hybrid approach: apply the rules deterministically + train the model to guess only once the simple logic code gets stuck? Presumably it would lower the required training set, perhaps significantly, and leverage the strength of each part.
This can better separate the 2 functionalities that the model learns simultaneously.

Have you tried rephrasing the training setting (and loss) in terms of standard RL metrics? The “moves” of the game are the guess steps, the reward (win/lose) is simply whether the board reaches an “e” or a “d”.
This is not mandatory of course - just seems to me like a more natural setting than learning transcripts of solutions.

**Ethical Concerns:**

["NO or VERY MINOR ethics concerns only"]

**Final Justification:**

This is a technically robust and well written work, potentially applicable to a variety of "hash-like" problems.

The main strength in my view is the rigorous testing.

I maintain my original recommendation to accept.

**Limitations:**

Yes

**Quality:**

3

**Strengths And Weaknesses:**

Strengths:

A solid work - in terms of engineering, experimental design, simplicity of approach, and efficiency.

Well written and clearly explained (even if a few concepts in the intro become clear only later on), with high quality figures 2 and 5.
SOTA on Sudoku benchmarks.

Potentially applicable to a wide range of puzzles and mathematical challenges - even broader than the authors mention explicitly.

Weaknesses:

The way I understood the system, there are 2 main parts: teaching the transformer the 4 basic Sudoku rules - imitating a deterministic algorithm - and teaching to make good guesses, specifically such that a single guess will bring to a full solution. While the second part can naturally leverage the flexible nature of networks, the first part seems more like a test of the tokenization and dataset generation choices - not a use case that would naturally benefit from DL.

The authors apply their method to other NP satisfiability problems to show generalization - specifically to 1-in-3 SAT, but it is only stated in the main paper - no details or description given. The information is in Appendix G (incorrectly called H in line 176).

While LLMs are mentioned as motivation and potential impact of the paper (both in the intro and summary), this system resembles more an RL setting, or maybe action/time series data. I believe this approach is useful even without connection to LLMs (there are many transformer systems that have nothing to do with natural language) - but the claim should either be strengthened or removed.

---

Line 155: *in

Line 310: “that the L1 is theoretically” - “the” should be removed.

---

> ### Author Rebuttal · Authors · 2025-07-31
>
> We appreciate the Reviewer's time and effort in reviewing our work. The feedback has highlighted important aspects for clarification, and we hope that further discussion will help address any remaining concerns. Below, we provide their responses to each point:
> 1) In our work we focused on training Transformers for combinatorial problems with problem-specific tokens rather than free-form text. However, our pipeline is based only on decoder-only Transformers and can be directly combined with natural language expressing the transcripts in plain English - instead of using token 134 we could instead write "I place number 4 in row 1 and column 3". We opted for a more succinct representation for efficiency in training and evaluation.
>
>     As we discuss in our answer to Reviewer 8BxY, there are additional opportunities for applying our framework to other problems, including mathematics as you already suggested.
>
> 2) Due to page limitations, we decided to add the detailed results and descriptions of the SAT experiments in Appendix. If the paper is accepted, we plan to extend the main manuscript to include further details on the SAT problem.
>
> 3) We have corrected the typos mentioned.
>
> 4) Sorry for the confusion regarding the oracle terminology and we will clarify further in the paper to make it clear.
>
>     We provide two theoretical oracles as a reference to help parse the results, one upper-bound and one lower-bound, and are unrelated to transformers or specific architectures.
>
>     The upper-bound oracle corresponds to knowing the full Sudoku solution but not which cells are backdoors. This corresponds to guessing the correct value for a random unfilled cell (after the basic rules are applied) but this guess may not be a backdoor resulting in a restart.
>
>     The lower-bound is similar but does not assume we know the solution.  After the basic rules are applied, it chooses a uniformly random (cell, value) combination that does not conflict with the existing values, i.e. appears valid but may not be correct and restarts if the combination is not a backdoor. Note that a backdoor must necessarily be correct as the solution to a Sudoku is always unique.
>
> 5) Indeed, the number of puzzles seen in the r,c,v representation (shown in Figure 3) is 3 times lower. In our work we do not consider the number of data-points as we can produce novel puzzles using our Random Puzzle Generator. It seems that the important parameter for improving accuracy is the number of tokens consumed by the transformer as its convergence is slow in the initial phase. For a fair comparison, we compared the number of tokens consumed which also accurately reflects the wall-clock training time as the models are identical. We can thus observe that our rcv representation can achieve higher accuracy compared to r,c,v for the same training time. Also r,c,v needs much more training time to achieve same accuracy as the rcv approach.
>
> 6) The "backdoor property" we refer to is primarily an empirical observation, 99,8\% of the Sudoku Puzzles from our Random Generator can be solved using at most 1 Guess (backdoor cell). This holds for our chosen and well-defined distribution of random Sudoku Puzzles. Since it is difficult to find which is the right backdoor, our model requires more guesses to identify this.
>
>     As shown in Figure 6 and discussed in lines 318–323, for more than 50\% puzzles typically 1-2 tries are sufficient to identify the backdoor. We reworded this part of the manuscript to make it clear that this is an empirical finding rather than a theoretically established property of all Sudoku instances.
>
> 7) We have not explored this hybrid approach, as our primary goal was to demonstrate that vanilla Transformers can acquire a generalizable and interpretable reasoning strategy. Note that our model can still be directly used for inference in a manner to the hybrid approach you describe.
>
> We hope our responses help address the concerns you raised. If there are any remaining points that need clarification, please feel free to let us know, we would be happy to discuss them further.

---

> ### Comment · Reviewer_stHu · 2025-08-02
>
> 1) Yes - I understand the formatting could have been more “natural language-like”, but that’s exactly my point - it can also not be. Transformers existed before their application to LLMs, and I don’t see an advantage, or in fact any direct relation between your system to anything language-related.
> If you insist on creating this connection, then you need to compare performance of different models, with and without fine-tuning etc.
>
> 5) Using wall-clock measurement as the metric is completely legitimate, but it still looks like an advantage of the representations, not the system.
>
> 6) Yes I understood that it’s an empirical property - the question is whether your network learns to identify it faster (less attempts) with time. Also, I’m still not sure I understand why, during inference, you backtrack at the second failed attempt. During training - to train the network to guess the backdoor move - that I understand, but during inference it’s unclear to me.

---

> > ### Author Response · Authors · 2025-08-06
> >
> > Thank you for the follow-up, Reviewer stHu!
> >
> > 1. Indeed, we agree that Transformers are useful beyond language models. Our work offers novel methods for automatically solving combinatorial problems driven by data. For many NP-complete problems like SAT, researchers have developed a number of different heuristics that enable navigating more efficiently the space of all possible combinations to arrive at a solution. Our framework can be seen as a way for automatically learning the right guesses that lead to a solution. Similar AI based solutions have been explored in past work such as Learning to Branch by Balcan et al.
> > However, while we did not explore it fully in our work, our framework also has strong connections to language models. Our whole framework boils down to defining a custom loss over a standard decoder only transformer which can be used in tandem with other language-specific losses or during fine-tuning. These can make LLMs that currently fail to solve even simple Sudoku puzzles, to reason more effectively and consistently arrive at good solutions at least for the specific domain of combinatorial problems that they have been taught. It is an interesting question whether teaching this reasoning for multiple combinatorial problems at the same time can result in more generalizable logic for other combinatorial problems.
> >
> > 2.  We agree that some of the improvements on the training time come because of the more compact representation. Additional running time improvements are obtained by giving multi-label loss as opposed to the standard next-token prediction which is significantly more noisy. Yet, the most important aspect of the system comes from the refined backtracking transcripts that teach the model how to navigate the solution space effectively and the novel framing of the problem as a Min-Sum Set Cover that boosts the probability of the correct guesses that make the model arrive at a solution faster. These are aspects that are independent of the representation and could also apply in other representations such as natural language.
> >
> > 3. Good point and this is an important aspect of our work that makes our approach so effective. While the first part of the work does not take back the tokens and continues after a failed guess the second part retracts the tokens after a failure and randomly retries again. We think this is an important conceptual contribution that allows the reasoning model to learn much more effectively allowing accuracies 99.8%. Indeed with this method the state-space of the Transformer and its context is significantly reduced. Remembering the failed choices not only pollutes the context but also requires significantly more training. One would need to generate during training sequences illustrating the failed attempts so that the model knows how to deal with this additional context. Instead here, we can train the model only with successful guesses and potentially at most one failed attempt so that it knows when it fails. At a higher level, we use randomness instead of memory to speed up learning and achieve high performance results with less parameters and less training sequences. Randomization is a core idea in complexity theory which allows solving the reachability problem using logarithmic space.

---

> > > ### Comment · Reviewer_stHu · 2025-08-08
> > >
> > > Thank you for your comments.
> > >
> > > I still maintain that the connection to the broader domain of LLM's is strongly claimed, yet barely discussed - not to mention proven.
> > >
> > > A good example is, as you mention yourself, fine tuning an LLM on such tasks (even using your step encoding) - perhaps multiple puzzle types at once - and then measure their relative performance.

---

### Official Review · Reviewer_8BxY · 2025-07-02

**Clarity:** 3
**Significance:** 2
**Originality:** 3
**Rating:** 4
**Confidence:** 4

**Summary:**

This paper presents a framework to teach Transformers to solve combinatorial problems, using Sudoku as the primary testbed. The method combines imitation learning (of simple rule-based logic) with search, using Depth First Search informed guessing and backtracking. The authors introduce action-level tokenization and multi-target training strategy, achieving 99% accuracy on Sudoku and generalizing to 1-in-3 SAT which is an NP hard problem. The second contribution is a theoretical and empirical analysis of minimizing solution length by formulating the guessing optimization as a contextual Min-Sum Set Cover problem, resulting in a new loss function that outperforms baseline cross-entropy loss.

**Questions:**

How does the proposed framework perform on Sudoku-Bench?

The DFS-based backtracking search is hard to scale. How does the model perform on larger Sudoku grids or harder SAT variants? Are there known scalability bottlenecks? Is there any further optimization needed?

Have the authors studied the impact of allowing deeper DFS (beyond depth-1)? What is the trade-off in performance and runtime?

**Ethical Concerns:**

["NO or VERY MINOR ethics concerns only"]

**Final Justification:**

Thanks for clarification. I will raise the score

**Limitations:**

Yes.

**Paper Formatting Concerns:**

NA.

**Quality:**

2

**Strengths And Weaknesses:**

1) The proposed method is well designed, clearly described, and simple.
2) This paper explores the use of LLMs for combinatorial reasoning, showing that trial-and-error with backtracking can be learned effectively.
3) The reframing of guessing optimization as a contextual Min-Sum Set Cover problem (and the new loss function as a result) is interesting.

Weakness:
1) The DFS-based backtracking search may not scale to harder combinatorial problems or larger sizes without further optimization. Authors didn't discuss the possible solutions to address problem size.

2) Evaluation is only limited to Sudoku and 1-in-3 SAT. The paper claims the effectiveness of using the method on general NP problems, but fails to provide more evaluations to support this claim.

3) This paper has missed discussing and comparing important related work such as: "Neural Combinatorial Optimization with Reinforcement Learning" Bello et al., 2016 and the many follow up works that came out afterwards.

Overall, The DFS-based backtracking search is an effective way to solve general NP problems, however, the scalability of the method on larger problems is challenging. This paper presented a way to integrate imitation learning of simple rule-based logic with Depth First Search (DFS) informed guessing and backtracking. The proposed optimization framework is limited to depth-1 in DFS and non-adaptive guessing. This will limit the application of this work on general NP problems. Moreover, the paper claimed to be effective to solve general NP problems, but failed to provide diverse evaluation to support this claim.

---

> ### Author Rebuttal · Authors · 2025-07-31
>
> We sincerely appreciate the Reviewer's insightful review of our work which have enabled us to address the issues, and we hope that through further discussion, we can clarify any remaining concern. Below, we provide our responses to each point:
> 1) We agree that a naive DFS approach can be intractable for large-scale combinatorial problems. However, the idea of combining DFS with simple rules, like DPLL, is quite powerful and is known to solve even large scale combinatorial problems like SAT. Our first finding is that even small Transformers can learn to implement such rules with DFS.
>
>     Once we realize that Transformers, are capable of reliably solving hard combinatorial problems, we also realize that doing it efficiently is paramount.
>
>     The key advantage of our framework is the integration of clever guessing (see Figures 2 and 5).
>
>     Even plain DFS can be made faster by rewarding correct guesses instead of just following the DFS rules. Our work in the second part of the paper illustrates a more clever pipeline that rewards guesses that are simultaneously correct and useful and explicitly limits the depth of search. This pipeline is significantly more efficient and exploits the powerful predictive capabilities of Transformers.
>
>     We formulated the guess selection problem as a Min-Sum Set Cover problem, which reduces the number of possible paths to explore and speeds up the solving process (see also Section 3 the losses comparison).
>
> 2) Our framework applies to any problem in NP and while we only illustrate this for Sudoku and 1-in-3 SAT, it is really easy to integrate any other NP problem. Note that since 1-in-3 SAT is NP-complete, by the Cook-Levin theorem, any problem in NP can be written as 1-in-3 SAT.
>
>     However, some problems may benefit from problem-specific optimizations and therefore, we provide a library in Appendix F2 that lets one generate transcripts for any problem in NP by providing the appropriate components.
>
>     Our framework has essentially 3 components that are learnable by a transformer. Solution generation, Solution verification and Guess generation (Hints).
>
>     For any problem in NP, these components are simple to obtain. Problems in NP have short certificates meaning that few guesses are needed to arrive at a solution. Moreover, problems in NP have efficient solution verification meaning that it is easy to verify correct solutions. The challenging part for NP (or NP-complete) problems is to arrive at a good solution and it is an open problem whether this can be done efficiently. As we discussed in our previous response, our goal is to optimize efficiency by automatically learning correct and useful guesses.
>
>     While our exploration was limited to problems in NP, our framework can also apply in other domains, as reviewer stHu mentioned, with efficient verification like verification of mathematical proofs or program correctness. In fact a similar pipeline was used in the recent work https://arxiv.org/abs/2507.15855 for solving math problems for IMO 2025
>     where hints like "mathematical induction" and "analytic geometry" where given to a solver that produced proofs that were subsequently verified using an LLM-based proof verifier.
>
>
> 3) We are familiar with the work of Bello et al. (2016) and follow-up works and will add a short discussion in the paper.
>
> 4) While we have not looked at Sudoku-Bench which seems to be a newer dataset and has been released after our submission, there are a number of Sudoku benchmarks that exist of varying difficulty. Our goal was to provide a comprehensive benchmark and Sudoku generator which is transparent in its puzzle selection which is essentially uniformly random Sudoku boards and thus not biased towards a particular solution.
>
>
> 5) Thank you for your question regarding the depth of guesses.
>
>     We note that our first part based on imitation learning does not have any constraints in depth and is very effective getting 99\% accuracy. One can allow arbitrary depths or use DFS with iterative deepening until a solution is found.
>
>     What we have not explored yet and it is an interesting direction for follow-up work is how to optimize multi-level guesses to minimize time-to-solution. We describe this in more detail in the section - Summary and Future Directions. While the single level DFS that we present in this work was already sufficient for the challenging task of Sudoku, optimizing multiple level guesses is important for other problems. While the 1-level guess corresponds to Min-Sum Set Cover which is well studied, optimizing multi-level guesses corresponds to the Min-Sum Group-Steiner Tree problem and requires additional insights (theoretical and applied) to integrate in the training pipeline that we plan to explore in future work.
>
>
> We hope these answers clarify the points you raised. Please let us know if there is anything else we should elaborate on, we would be glad to discuss further.

---

> > ### Comment · Reviewer_8BxY · 2025-08-05
> >
> > Thanks for clarification. I will raise the score.

---

> > > ### Author Response · Authors · 2025-08-06
> > >
> > > We sincerely appreciate your careful review of our work Reviewer 8BxY and your decision to raise the score!

---

### Note · Authors · 2025-08-15

Dear All,

We thank the reviewers for their careful and constructive feedback and are pleased that our rebuttal responses were well received. We have incorporated all suggestions into the revised draft, clarifying our contributions and emphasizing that our work achieves state-of-the-art results on the challenging reasoning task of Sudoku. More broadly, our framework offers a generalizable approach for solving any NP problem with Transformers, which we believe opens up promising new research directions. For example, in the current version, our decoder-only architecture can be adapted to natural language transcripts; exploring free-form natural language integration remains an interesting future direction, as noted in our revised conclusion.

We thank again all the reviewers for their time and valuable feedback.

---

### Decision · Program_Chairs · 2025-09-17

**Decision:**

Accept (poster)

**Comment:**

In this paper the authors propose a new framework for solving combinatorial problems such as Sudoku using a standard decoder-only Transformer model. The key idea is to combine supervised imitation learning of basic Sudoku rules with a trial-and-error Depth-First Search (DFS) strategy to make informed guesses and backtrack when encountering conflicts. This leads to solving problems that cannot be addressed by heuristic rule application alone. In addition, the authors design a loss function inspired by the Min-Sum Set Cover problem to optimize inference efficiency. The proposed method achieves near-perfect accuracy (99%) on Sudoku and 1-in-3 SAT tasks, outperforming prior neural and neuro-symbolic baselines.

The key strength of the paper is the fact that they can teach their model to do backtracking. On the other hand, the claims of the paper seems to be a bit of oversell as their work has only been tested on a small set of problems.

There has been some discrepancy between the authors regarding the novelty and significance of this paper. However, I strongly believe that the backtracking contribution is impressive and therefore it is worth presenting this paper at the conference.